# Near-Optimal Adversarial Reinforcement Learning with Switching Costs

**Ming Shi, Yingbin Liang, Ness Shroff**
Department of Electrical and Computer Engineering
The Ohio State University
Columbus, OH 43210, USA
`{shi.1796,liang.889,shroff.11}@osu.edu`

## Abstract

Switching costs, which capture the costs for changing policies, are regarded as a critical metric in reinforcement learning (RL), in addition to the standard metric of losses (or rewards). However, existing studies on switching costs (with a coefficient that is strictly positive and is independent of the time horizon) have mainly focused on static RL, where the loss distribution is assumed to be fixed during the learning process, and thus practical scenarios where the loss distribution could be non-stationary or even adversarial are not considered. While adversarial RL better models this type of practical scenarios, an open problem remains: how to develop a provably efficient algorithm for adversarial RL with switching costs? This paper makes the first effort towards solving this problem. First, we provide a regret lower-bound that shows that the regret of any algorithm must be larger than $\tilde{\Omega}((HSA)^{1/3}T^{2/3})$, where $T$, $S$, $A$ and $H$ are the number of episodes, states, actions and layers in each episode, respectively. Our lower bound indicates that, due to the fundamental challenge of switching costs in adversarial RL, the best achieved regret (whose dependency on $T$ is $\tilde{O}(\sqrt{T})$) in static RL with switching costs (as well as adversarial RL without switching costs) is no longer achievable. Moreover, we propose two novel switching-reduced algorithms with regrets that match our lower bound when the transition function is known, and match our lower bound within a small factor of $\tilde{O}(H^{1/3})$ when the transition function is unknown. Our regret analysis demonstrates the near-optimal performance of them.

## 1 Introduction

Reinforcement learning (RL) recently arises as a compelling paradigm for modeling machine learning applications with sequential decision making. In such a problem, an online learner interacts with the environment sequentially over Markov decision processes (MDPs), and aims to find a desirable policy for achieving an accumulated loss (or reward). Various algorithms have been developed for RL problems and have been shown theoretically to achieve polynomial sample efficiency in Zimin & Neu (2013); Azar et al. (2017); Jin et al. (2018); Agarwal et al. (2019); Bai et al. (2019); Jin et al. (2020a;b); Cai et al. (2020); Gao et al. (2021); Lykouris et al. (2021); Qiao et al. (2022), etc.

In addition to the metric of losses, **switching costs**, which capture the costs for changing policies during the execution of RL algorithms, are also attracting increasing attention. This is motivated by many practical scenarios where the online learners cannot change their policies for free. For example, in recommendation systems, each change of the recommendation involves the processing of a huge amount of data and additional computational costs (Theocharous et al., 2015). Similarly, in healthcare, each change of the medical treatment requires substantial human efforts and time-consuming tests and trials (Yu et al., 2021). Such switching costs are also required to be considered in many other areas, e.g., robotics applications (Kober et al., 2013), education software (Bennane, 2013), computer networking (Xu et al., 2018), and database optimization (Krishnan et al., 2018).

Switching costs have been studied in various problems (please see Sec. 2 for some examples). Among these studies, a relevant line of research is along bandit learning (Geulen et al., 2010; Dekel et al., 2014; Arora et al., 2019; Shi et al., 2022). Recently, switching costs have received consider-

able attention in more general RL settings (Bai et al., 2019; Gao et al., 2021; Wang et al., 2021; Qiao et al., 2022). However, these studies have mainly focused on *static* RL, where the loss distribution is assumed to be fixed during the learning process. Thus, practical scenarios where the loss distribution could be non-stationary or even adversarial are not characterized or considered.

While **adversarial RL** better models the non-stationary or adversarial changes of the loss distribution, to the best of our knowledge, an open problem remains: *how to develop a provably efficient algorithm for adversarial RL with switching costs?* Intuitively, in adversarial RL, since much more often policy switches would be needed to adapt to the time-varying environment, it would be much more difficult to achieve a low regret (including both the standard loss regret and the switching costs, please see (6)). Indeed, without a special design to reduce switching, existing algorithms for adversarial RL with $T$ episodes, such as those in Zimin & Neu (2013); Jin et al. (2020a); Lee et al. (2020) and Lykouris et al. (2021), could yield poor performance with linear-to-$T$ number of policy switches. *Thus, the goal of this paper is to make the first effort along this open direction.*

Our first aim is to develop provably efficient algorithms that enjoy low regrets in adversarial RL with switching costs. This requires a careful reduction of switching under non-stationary or adversarial loss distributions. It turns out that previous approaches to reduce switching in *static* RL (e.g., those in Bai et al. (2019) and Qiao et al. (2022)) are not applicable here. Specifically, the high-level idea in static RL is to switch faster at the beginning, while switch slower and slower for later episodes. Such a method performs well in static RL, mainly because after learning enough information about losses at the beginning (by switching faster), the learner can estimate the assumed *fixed* loss-distribution accurately enough with high probability in later episodes. Thus, even though the learner switches slower and slower, a low regret is still achievable with high probability. In contrast, when the loss distribution could change arbitrarily, this method does not work. This is mainly because what the learner learned in the past may not be that useful for the future. For example, when the loss distribution is adversarial, a state-action pair with small losses in the past may incur large losses in the future. Thus, new ideas are required for addressing switching costs in adversarial RL.

Our second aim is to understand fundamentally whether the new challenge of switching costs in adversarial RL significantly increases the regret. This requires a converse result, i.e., a lower bound on the regret, that holds for any RL algorithm. Further, we aim to understand fundamentally whether the adversarial nature of RL indeed requires much more policy switches to achieve a low loss regret.

**Our contributions:** In this paper, we achieve the aforementioned goals and make the following three main contributions. (We use $\tilde{\Omega}$, $\tilde{\Theta}$ and $\tilde{O}$ to hide constants and logarithmic terms.)

**First**, we provide a lower bound (in Theorem 1) that shows that, for adversarial RL with switching costs, the regret of any algorithm must be larger than $\tilde{\Omega}((HSA)^{1/3}T^{2/3})$, where $T$, $S$, $A$ and $H$ are the number of episodes, states, actions and layers in each episode, respectively. Our lower bound indicates that, due to the fundamental challenge of switching costs in adversarial RL, the best achieved regret (whose dependency on $T$ is $\tilde{O}(\sqrt{T})$) in static RL with switching costs is no longer achievable. Further, we characterize precisely the new trade-off (in Theorem 2) between the standard loss regret and the switching costs due to the adversarial nature of RL.

**Second**, we develop the first-known near-optimal algorithms for adversarial RL with switching costs. As we discussed above, the idea for reducing switching in static RL does not work well here. To handle the losses that can change arbitrarily, our design is inspired by the approach in Shi et al. (2022) for bandit learning, but with two novel ideas. (a) We delay each switch by a fixed (but *tunable*) number of episodes, which ensures that switch occurs only every $\tilde{O}(T^{1/3})$ episodes. (b) The idea in (a) results in consistently long intervals of not switching. Since the bias in estimating losses from such a long interval tends to increase the regret, it is important to construct an unbiased estimate of losses for each interval. To achieve this, the idea in bandit learning is to consider all time-slots in each interval as one time-slot, which necessarily requires a *single* chosen action in each interval. Such an approach is not applicable to our more general MDP setting, since there is no guarantee to visit a *single* state-action pair due to state transitions. To resolve this issue, our novel idea is to decompose each interval, and then combine the losses of each state-action pair only from the episodes in which such a state-action pair is visited. Interestingly, although this combination is random and the loss is adversarial, the expectation of the estimated losses is (almost) unbiased.

**Third**, we establish the regret bounds for our new algorithms. For the case with a *known* transition function, we show that our algorithm achieves an $\tilde{O}((HSA)^{1/3}T^{2/3})$ regret, which matches our lower bound. For the case with an *unknown* transition function, we show that, with probability $1 - \delta$, our algorithm achieves an $\tilde{O}\left(H^{2/3}(SA)^{1/3}T^{2/3}(\ln\frac{TSA}{\delta})^{1/2}\right)$ regret, which matches our lower bound on the dependency of $T$, $S$ and $A$, except with a small factor of $\tilde{O}(H^{1/3})$. Therefore, the regrets of our new algorithms are near-optimal. Moreover, because of our novel ideas for estimating losses and delaying switching discussed above in a state-transition case, our proofs for the regrets involve several new analytical ideas. For example, in Lemma 1 and Lemma 2, we show that our new way of estimating losses is (almost) unbiased so that its effect on the regret is controllable. Moreover, to capture the effects of the delayed switching, our new analytical idea is to first bound the regret across intervals between adjacent switching events, and then relate the regret inside episodes of each interval to this bound (please see *Step-2* of the proofs in Appendix D and Appendix G).

## 2 RELATED WORK

**Switching costs:** Switching costs have already received considerable attention in various online problems. For example, online convex optimization with switching costs has been studied in Lin et al. (2012); Chen et al. (2016); Goel et al. (2019); Shi et al. (2021a;b), etc. Convex body chasing with switching costs has been studied in Friedman & Linial (1993); Sellke (2020); Bubeck et al. (2021), etc. Switching costs have also been studied in metrical task systems (Borodin & El-Yaniv, 2005), online set covering (Buchbinder et al., 2014), $k$-server problem (Lin et al., 2020), online control (Goel & Wierman, 2019; Li et al., 2020; Lin et al., 2021), etc. Moreover, switching costs have been studied in adversarial bandit learning, e.g., in Geulen et al. (2010); Dekel et al. (2014); Arora et al. (2019); Shi et al. (2022). Our work in this paper can be viewed as a non-trivial generalization of these studies on bandit learning to adversarial MDP, where state transitions and multiple layers in each episode require new developments in both the algorithm design and regret analysis.

**Static MDP:** There have been recent studies on static RL with switching costs. Specifically, for tabular MDP, Bai et al. (2019) and Zhang et al. (2020) proposed RL algorithms that attain an $\tilde{O}\left(\sqrt{H^\alpha SAT} \cdot \ln\frac{TSA}{\delta}\right)$ regret with probability $1 - \delta$, by incurring $O\left(H^\alpha SA \ln T\right)$ switching costs, where $\alpha = 3$ and 2, respectively. Recently, Qiao et al. (2022) obtained a similar $\tilde{O}(\sqrt{T})$ regret with probability $1 - \delta$, by incurring $O\left(HSA \ln\ln T\right)$ switching costs. Moreover, for linear MDP (with $d$-dimensional feature space), Gao et al. (2021) and Wang et al. (2021) obtained an $\tilde{O}\left(\sqrt{d^3 H^3 T} \cdot (\ln\frac{dT}{\delta})^{1/2}\right)$ regret with probability $1 - \delta$, by incurring $O\left(dH \ln T\right)$ switching costs.

**Adversarial MDPs:** Adversarial RL better models scenarios where the loss distributions and/or the transition functions of MDPs could change over time. Specifically, in tabular MDP with a known transition function, Zimin & Neu (2013) proposed an RL algorithm that attains an $\tilde{O}(\sqrt{HSAT})$ regret. In the case with an unknown transition function, Jin et al. (2020a) and Lee et al. (2020) obtained an $\tilde{O}\left(HS\sqrt{AT \ln\frac{TSA}{\delta}}\right)$ regret with probability $1 - \delta$. These studies assume that the state spaces of layers in an episode are non-overlapping. Moreover, Rosenberg & Mansour (2019a) studied the case with full-information feedback. Adversarial linear MDP has also been studied recently, e.g., in Cai et al. (2020); Luo et al. (2021). In addition, Yu & Mannor (2009); Cheung et al. (2019) and Lykouris et al. (2021) studied the case when both the loss distribution and transition function change arbitrarily. More studies on various adversarial RL settings have been done by Rosenberg & Mansour (2019b); Lee et al. (2021); Zhao et al. (2021); Jin et al. (2021); He et al. (2022), etc.

To the best of our knowledge, no study in the literature has addressed the challenge due to *switching costs in adversarial RL*, which is the focus of this paper.

## 3 PROBLEM FORMULATION

We consider adversarial reinforcement learning (RL) with switching costs in episodic Markov decision processes (MDPs). Suppose there are $T$ episodes, each of which consists of $H$ layers. We use $\mathcal{S}_h$ to denote the state space of layer $h$. For ease of elaboration, as in previous work (e.g., Zimin & Neu (2013); Jin et al. (2020a) and Lee et al. (2020)), we assume that the $H$ layers are non-

intersecting, i.e., $\mathcal{S}_{h'} \cap \mathcal{S}_{h''} = \phi$ for any $h' \neq h''$; $\mathcal{S}_0 = \{s_0\}$ is a singleton; and each episode ends at state $\mathcal{S}_H = \{s_H\}$. Thus, the entire state space is $\mathcal{S} = \cup_{h=0}^{H} \mathcal{S}_h$ with size $S = \sum_{h=0}^{H} S_h$, where $S_h$ denotes the size of $\mathcal{S}_h$. Moreover, we use $\mathcal{A}$ to denote the action space with size $A$. Then, the MDP is defined by a tuple $\left(\mathcal{S}, \mathcal{A}, P, \{l_t\}_{t=1}^{T}, H\right)$, where $P$ is the transition function with $P_h : \mathcal{S}_{h+1} \times \mathcal{S}_h \times \mathcal{A} \to [0,1]$ denoting the transition probability measure at layer $h$, and $l_t : \mathcal{S} \times \mathcal{A} \to [0,1]$ represents the loss function for episode $t$.

The online learner interacts with the Markov environment episode-by-episode as follows. At the beginning of each episode $t = 1, ..., T$, the online learner starts from state $s_0$ and follows an algorithm that (possibly randomly) chooses a *deterministic* policy $\pi_t : \mathcal{S} \to \mathcal{A}$. Next, at each layer $h = 0, ..., H-1$, after observing the current sate $s_{t,h}$, the learner chooses an action $a_{t,h} = \pi_t(s_{t,h})$. Then, the learner incurs a loss $l_t(s_{t,h}, a_{t,h})$. Finally, the next state $s_{t,h+1} \in \mathcal{S}_{h+1}$ is drawn according to the transition probability $P(\cdot|s_{t,h}, a_{t,h})$. (For simplicity, we drop the index $h$ of $P_h$ in this paper when it is clear from the context.) These steps repeat until the learner arrives at the last state $s_H$. At the end of episode $t$, only the losses of visited state-action pairs in the episode are observed by the learner, whereas the losses of non-visited state-action pairs are unknown. As in (Zimin & Neu, 2013; Jin et al., 2020a; Lee et al., 2020; Cai et al., 2020), this is called "**bandit feedback**", which is more practical than full-information feedback (Rosenberg & Mansour, 2019a) that assumes the losses of all state-action pairs (no matter visited or not) are known for free.

**Adversarial losses:** Different from static RL that assumes the loss distribution is fixed for all episodes, in the adversarial setting we consider here, we do not need any assumption on the underlying loss distribution. That is, the loss function $l_t$ could change arbitrarily across episodes.

**Switching costs:** As we mentioned in the introduction, in adversarial RL, addressing switching costs remains an open problem. The switching cost refers to the cost needed for changing the policy $\pi_t$. It is equal to $\beta \cdot \mathbf{1}_{\{\pi_{t+1} \neq \pi_t\}}$, where $\beta > 0$ is the switching-cost coefficient and is independent of $T$, and $\mathbf{1}_{\mathcal{E}}$ is an indicator function (i.e., $\mathbf{1}_{\mathcal{E}} = 1$ if the event $\mathcal{E}$ occurs, and $\mathbf{1}_{\mathcal{E}} = 0$ otherwise).

Therefore, the total cost of executing an RL algorithm $\pi$ over $T$ episodes is given by

$$\text{Cost}^\pi(1:T) \triangleq \mathbb{E}\left[\sum_{t=1}^{T}\sum_{h=0}^{H-1} l_t(s_{t,h}^\pi, a_{t,h}^\pi) + \sum_{t=1}^{T-1} \beta \cdot \mathbf{1}_{\{\pi_{t+1} \neq \pi_t\}} \Big| \pi, P\right], \qquad (1)$$

where the expectation is taken with respect to the randomness of the state-action pairs $(s_{t,h}^\pi, a_{t,h}^\pi)$ visited by $\pi$, and the possible randomness of changing the policy $\pi_t$.

Next, we introduce a concept called "occupancy measure" (Zimin & Neu, 2013; Jin et al., 2020a). Specifically, the occupancy measure $q_t^{\pi,P}(s,a) = Pr[s_{t,h}^\pi = s, a_{t,h}^\pi = a|\pi, P] \geq 0$ is the probability of visiting the state-action pair $(s,a)$ by the algorithm $\pi$ at layer $h$ of episode $t$ under the transition function $P$. In addition (with slight abuse of notation), the occupancy measure $q_t^{\pi,P}(s',s,a) = Pr[s_{t,h+1}^\pi = s', s_{t,h}^\pi = s, a_{t,h}^\pi = a|\pi, P] \geq 0$ is the probability of visiting the state-action triple $(s',s,a)$ by the algorithm $\pi$ at layers $h$ and $h+1$ of episode $t$ under the transition function $P$. In order to be feasible, the occupancy measures need to satisfy some conditions at layer $h$ of episode $t$. First, according to probability theory, they need to satisfy the conditions that,

$$q_t^{\pi,P}(s,a) = \sum_{s' \in \mathcal{S}_{h+1}} q_t^{\pi,P}(s',s,a), \text{ for all } (s,a) \in \mathcal{S}_h \times \mathcal{A}, \text{ and } \sum_{s \in \mathcal{S}_h}\sum_{a \in \mathcal{A}} q_t^{\pi,P}(s,a) = 1. \quad (2)$$

Second, since the probability of transferring to a state $s$ from the previous layer $h-1$ must be equal to the probability of transferring from this state $s$ to the next layer $h+1$, we have

$$\sum_{s' \in \mathcal{S}_{h-1}}\sum_{a \in \mathcal{A}} q_t^{\pi,P}(s,s',a) = \sum_{s' \in \mathcal{S}_{h+1}}\sum_{a \in \mathcal{A}} q_t^{\pi,P}(s',s,a), \text{ for all } s \in \mathcal{S}_h. \qquad (3)$$

Third, the occupancy measure should generate the true transition function $P$, i.e.,

$$\frac{q_t^{\pi,P}(s',s,a)}{\sum_{b \in \mathcal{A}} q_t^{\pi,P}(s',s,b)} = P_h(s'|s,a), \text{ for all } (s',s,a) \in \mathcal{S}_{h+1} \times \mathcal{S}_h \times \mathcal{A}. \qquad (4)$$

We use $\mathbb{C}(P)$ to denote the set of all occupancy measures that satisfy conditions (2)-(4). Moreover, at the beginning of episode $t$, the algorithm $\pi$ associated with the occupancy measure $q_t^{\pi,P}$ chooses a *deterministic* policy $\pi_t$ by assigning an action $a \in \mathcal{A}$ to each state $s \in \mathcal{S}$ according to the probability

$$Pr[a|s] = \frac{q_t^{\pi,P}(s,a)}{\sum_{b \in \mathcal{A}} q_t^{\pi,P}(s,b)}. \qquad (5)$$

Then, it is not hard to show that the expected total loss, i.e., the first term in (1), can be expressed as $\text{loss}^{\pi}(1:T) \triangleq \mathbb{E}\left[\sum_{t=1}^{T}\langle q_t^{\pi,P}, l_t\rangle \middle| \pi, P\right]$. Finally, the regret of an RL algorithm $\pi$ is defined to be the sum of the loss regret $R_{\text{loss}}^{\pi}(T)$ and the switching costs of as follows:

$$R^{\pi}(T) \triangleq \underbrace{\max_{q \in \mathbb{C}(P)} \mathbb{E}\left[\sum_{t=1}^{T}\langle q_t^{\pi,P} - q, l_t\rangle \middle| \pi, P\right]}_{\text{loss regret: } R_{\text{loss}}^{\pi}(T)} + \underbrace{\mathbb{E}\left[\sum_{t=1}^{T-1} \beta \cdot \mathbf{1}_{\{\pi_{t+1} \neq \pi_t\}} \middle| \pi, P\right]}_{\text{switching costs}}. \tag{6}$$

Therefore, our goal in this paper is to design RL algorithms that achieve as low regret as possible against any possible sequence of loss functions $\{l_t\}_{t=1}^{T}$ and state transition function $P$.

## 4  A LOWER BOUND

In this section, we will develop a lower bound on the regret for adversarial RL with switching costs. Such a lower bound will quantify how difficult it is to control the regret with switching costs under adversarial RL. In Theorem 1 below, we provide this lower bound, the proof of which is given in Appendix A. (In Sec. 5 and Sec. 6, we will provide two near-optimal RL algorithms to achieve this lower bound.)

**Theorem 1.** *For adversarial RL with switching costs and $T \geq \max\{6H^2SA, \beta\}$, the regret of any RL algorithm $\pi$ can be lower-bounded as follows,*

$$R^{\pi}(T) \geq \tilde{\Omega}\left(\beta^{1/3}(HSA)^{1/3}T^{2/3}\right). \tag{7}$$

Theorem 1 shows that in adversarial RL with switching costs, the dependency on $T$ of the best achievable regret is at least $\tilde{\Omega}(T^{2/3})$. Thus, the best achieved regret (whose dependency on $T$ is $\tilde{O}(\sqrt{T})$) in *static* RL with switching costs (in Bai et al. (2019); Qiao et al. (2022), etc) as well as adversarial RL *without* switching costs (in Zimin & Neu (2013); Jin et al. (2018), etc) is no longer achievable. This demonstrates the fundamental challenge of switching costs in adversarial RL, and it is expected that new challenges will arise when developing provably efficient algorithms.

Further, in Theorem 2 below, we characterize precisely the new trade-off between the loss regret and switching costs defined in (6). The proof is provided in Appendix B. Intuitively, by switching more, the online RL algorithm can adapt more flexibly to the new information learned, and thus achieves a lower loss regret. On the other hand, if fewer switches are allowed, the online RL algorithm is less flexible to adapt to the new information learned, which will incur a larger loss regret.

**Theorem 2.** *For adversarial RL with switching costs, with the switching costs equal to $O\left(\beta \cdot \mathcal{N}^{swi}\right)$, the loss regret can be lower-bounded by $\tilde{\Omega}\left(\sqrt{\frac{HSA}{\mathcal{N}^{swi}}} \cdot T\right)$. Alternatively, to achieve a loss regret equal to $\tilde{O}\left(\sqrt{\frac{HSA}{\mathcal{N}^{swi}}} \cdot T\right)$, the switching costs incurred must be larger than $\Omega\left(\beta \cdot \mathcal{N}^{swi}\right)$.*

Theorem 2 provides an interesting and necessary trade-off between the loss regret and switching costs. We further elaborate this result in three cases. **First**, in order to achieve a loss regret $\tilde{O}(H\sqrt{SAT})$, Theorem 2 shows that the number of switches $\mathcal{N}^{swi}$ (and thus the switching costs incurred) must be linear in $T$, i.e., essentially switching at almost all episodes. This is consistent with the regret achieved in adversarial RL *without* switching costs, i.e., allowing switching linear-to-$T$ number of times for free. But our result further implies that, without linear-to-$T$ switches of the policy, it is impossible to achieve an $\tilde{O}(\sqrt{T})$ loss regret. **Second**, Theorem 2 shows that, if only a constant or $O(\ln \ln T)$ number of switches are allowed, the loss regret must be linear in $T$. In contrast, in *static* RL, an $\tilde{O}(\sqrt{T})$ loss regret is achieved with only $O(\ln \ln T)$ switches (Qiao et al., 2022). This indicates that the adversarial nature of RL necessarily requires significantly more policy switches to achieve a low loss regret. **Third**, Theorem 2 suggests that the loss regret and switching costs can be balanced at the order of $\tilde{O}\left(T^{2/3}\right)$. That is, to achieve the $\tilde{O}\left(T^{2/3}\right)$ loss regret, the switching costs incurred have to be $\tilde{\Omega}\left(T^{2/3}\right)$. This is consistent with Theorem 1, where the regret (including both the loss regret and switching costs) is lowered-bound by $\tilde{\Omega}\left(T^{2/3}\right)$.

---

**Algorithm 1** Switching rEduced EpisoDic relative entropy policy Search (SEEDS)

---

**Parameters:** $\eta = \tilde{\Theta}\left(\beta^{-1/3}H^{2/3}(SA)^{-1/3}T^{-2/3}\right)$ and $\tau = \tilde{\Theta}\left(\beta^{2/3}(HSA)^{-1/3}T^{1/3}\right)$.

**Initialization:** $Pr[a|s] = \frac{1}{A}$ for all $(s,a) \in \mathcal{S} \times \mathcal{A}$. Choose $\pi_{[1]}^{\text{SEEDS}}$ according to (5).

**for** $u = 1 : \lceil \frac{T}{\tau} \rceil$ **do**
  **for** $t = (u-1)\tau + 1 : \min\{u\tau, T\}$ **do**
    *Step 1:* Execute the updated policy $\pi_{[u]}^{\text{SEEDS}} = \pi^{\hat{q}_{[u]}^{\text{SEEDS}, P}}$.
  **end for**
  At the end of super-episode $u$,
  *Step 2:* Estimate the losses $\hat{l}_{[u]}^{\text{SEEDS}}(s,a)$ for all $(s,a)$ according to (8).
  *Step 3:* Update the occupancy measure $\hat{q}_{[u+1]}^{\text{SEEDS}, P}(s,a)$ according to (10). Update the deterministic policy $\pi^{\hat{q}_{[u+1]}^{\text{SEEDS}, P}}$ according to (5).
**end for**

---

# 5 THE CASE WHEN THE TRANSITION FUNCTION IS KNOWN

In this section, we study the case when the transition function is *known*, and we will further explore the more challenging case when the transition function is *unknown* in Sec. 6. We propose a novel algorithm (please see Algorithm 1) with a regret that matches the lower bound in (7). Our algorithm is called Switching rEduced EpisoDic relative entropy policy Search (SEEDS).

SEEDS is inspired by the episodic method in bandit learning (Shi et al., 2022). In bandit learning, the idea is to divide the time horizon into $\Theta(T^{2/3})$ episodes, and pull one *single* Exp3-arm in an episode. By doing so, the total switching cost is trivially $O(T^{2/3})$. Meanwhile, the loss regret in an episode is $\Theta(\eta \cdot (T^{1/3})^2)$, which is proportional to the loss variance in an episode. The final $O(T^{2/3})$ regret is then achieved by taking the sum of all these costs and tuning the parameter $\eta = \Theta(T^{-2/3})$. However, in the adversarial MDP setting that we consider, there is a key difference due to random state-action visitations that cause several new challenges as we discuss in the rest of this section.

**Super-episode-based policy search:** SEEDS divides the episodes into $\mathcal{U} = \lceil \frac{T}{\tau} \rceil$ super-episodes, where $\tau \in \mathbb{Z}_{++}$ is a tunable parameter and a strictly positive integer. Each super-episode includes $\tau$ consecutive episodes. For all episodes in each super-episode $u = 1, ..., \mathcal{U}$, SEEDS uses the same policy $\pi^{\hat{q}_{[u]}^{\text{SEEDS}, P}}$ (*Step-1* in Algorithm 1) that was updated at the end of the last super-episode $u-1$, where $\hat{q}_{[u]}^{\text{SEEDS}, P}$ is the updated occupancy measure (that we will introduce soon) of SEEDS for super-episode $u$. Thus, SEEDS switches the policy at most once in each super-episode.

**A novel idea for estimating the losses:** At the end of super-episode $u$, SEEDS estimates the losses $l_{[u]}(s,a)$ of all state-action pairs in super-episode $u$. Here, it is instructive to see why the episodic importance-estimating method in adversarial bandit learning (i.e., without state transitions) does not apply to our problem. Note that due to state transitions in our more general MDP setting, we are not guaranteed to visit a *single* state-action pair for the whole super-episode. A naive but intuitive solution may be pretending that each state-action pair visited in super-episode $u$ was the *single* one visited. Then, we can let the estimated loss of each state-action pair $(s,a)$ to be $\hat{l}_{[u]}(s,a) = \frac{\bar{l}_{[u]}(s,a)}{1-(1-\hat{q}_{[u]}^{\text{SEEDS}, P}(s,a))^{\tau}} \mathbf{1}_{\{(s,a) \text{ was visited in super-episode } u\}}$, where the numerator $\bar{l}_{[u]}(s,a) = \sum_{t=(u-1)\tau+1}^{u\tau} l_t(s,a)/\tau$ is the average loss of $(s,a)$. If we assume that the loss $l_t$ for all episodes $t$ in super-episode $u$ were the same, according to the analysis in bandit learning and the inequality $1 - (1-x)^{\tau} \geq x$ for all $0 \leq x \leq 1$, this idea would have worked. However, the problem is that, inside super-episode $u$, the loss function $l_t$ for each episode $t$ could change arbitrarily. Thus, the estimated loss $\hat{l}_{[u]}(s,a)$ above is actually unknown and an ill-defined value.

To resolve the aforementioned difficulty due to randomly-visited state-action pairs and arbitrarily-changing loss functions, SEEDS estimates the loss as follows (*Step-2* in Algorithm 1),

$$\hat{l}_{[u]}^{\text{SEEDS}}(s,a) = \sum_{j=1}^{J_{[u]}} \frac{l_{t_j(s,a)}(s,a)}{\hat{q}_{[u]}^{\text{SEEDS}, P}(s,a)} \mathbf{1}_{\{(s,a) \text{ was visited in episodes } t_1(s,a), \ldots, t_{J_{[u]}}(s,a) \text{ of super-episode } u\}}, \quad (8)$$

where $J_{[u]}$ is the maximum number of episodes that the state-action pair $(s, a)$ was visited in super-episode $u$. In other words, in super-episode $u$, this state-action pair $(s, a)$ was not visited in any other episode $t$, such that $t \in \{(u-1)\tau + 1, ..., u\tau\}/\{t_1(s, a), ..., t_{J_{[u]}}(s, a)\}$. Thus, SEEDS estimates the losses based on the observable true losses in super-episode $u$. In this way, SEEDS elegantly resolves the aforementioned difficulty due to the random state transitions and adversarial losses. Our novel idea in (8) may be of independent interest for other problems with state transitions and non-stationary or adversarial losses. Indeed, in Sec. 6, we will apply this idea to the case when the transition function is unknown.

In Lemma 1 below, we show that the estimated loss in (8) is an unbiased estimation of the true loss in super-episode $u$. This is an important property that we will exploit in our regret analysis. The proof of Lemma 1 is provided in Appendix C. We use $\mathcal{F}_{[u]}$ to denote the $\sigma$-algebra generated by the observation of SEEDS before super-episode $u$.

**Lemma 1.** *The conditional expectation of the estimated loss designed in (8) is equal to*

$$\mathbb{E}\left[\hat{l}_{[u]}^{SEEDS}(s, a)\Big|\mathcal{F}_{[u]}\right] = l_{[u]}(s, a), \text{ for all } (s, a), \tag{9}$$

*where the expectation is taken with respect to the randomness of the episodes $t_1(s, a), ..., t_{J_{[u]}}(s, a)$, in which the state-action pair $(s, a)$ was visited, and $l_{[u]}(s, a) = \sum_{t=(u-1)\tau+1}^{\min\{u\tau, T\}} l_t(s, a)$ is the true loss of $(s, a)$ in super-episode $u$.*

**Updating the occupancy measure:** Finally, according to online mirror descent (Rakhlin et al., 2009; Zimin & Neu, 2013), SEEDS updates the occupancy measure $\hat{q}_{[u+1]}^{SEEDS,P}(s, a)$ for all state-action pairs $(s, a) \in \mathcal{S} \times \mathcal{A}$ as follows (*Step-3* in Algorithm 1),

$$\hat{q}_{[u+1]}^{SEEDS,P} = \arg\min_{q \in \mathbb{C}(P)} \left\{\eta \cdot \left\langle q, \hat{l}_{[u]}^{SEEDS}\right\rangle + D_{KL}\left(q \left\| \hat{q}_{[u]}^{SEEDS,P}\right.\right)\right\}, \tag{10}$$

where $D_{KL}(q\|q') \triangleq \sum_{s \in \mathcal{S}, a \in \mathcal{A}} q(s, a) \ln \frac{q(s,a)}{q'(s,a)} - \sum_{s \in \mathcal{S}, a \in \mathcal{A}} [q(s, a) - q'(s, a)]$ is the unnormalized relative entropy between two occupancy measures $q$ and $q'$ on the space $\mathcal{S} \times \mathcal{A}$. Recall that $\mathbb{C}(P)$ is formulated by (2)-(4). Note that the term $\langle q, \hat{l}_{[u]}^{SEEDS}\rangle$ represents the expected loss in super-episode $u$, with respect to the newly-estimated loss function $\hat{l}_{[u]}^{SEEDS}$. Thus, it captures how SEEDS adapts to and explores the newly-estimated loss function. In addition, the term $D_{KL}(q\|\hat{q}_{[u]}^{SEEDS,P})$ serves as a regularizer to ensure that the updated occupancy measure in (10) stays close to $\hat{q}_{[u]}^{SEEDS,P}$. Thus, it captures how SEEDS exploits the previously-estimated loss functions before super-episode $u$. As a result, by tuning the parameter $\eta$ in (10), the updated occupancy measure strikes a balance between exploration and exploitation.

We characterize the regret of SEEDS in Theorem 3 below.

**Theorem 3.** *Consider adversarial RL with switching costs introduced in Sec. 3. When the transition function $P$ is known, the regret of SEEDS is upper-bounded as follows,*

$$R^{SEEDS}(T) \leq \tilde{O}\left(\beta^{1/3}(HSA)^{1/3}T^{2/3}\right). \tag{11}$$

Theorem 3 shows that the regret of SEEDS matches the lower bound in (7) in terms of the dependency on all the parameters $T$, $S$, $A$, $H$ and $\beta$. Thus, the regret of SEEDS is order-wise optimal. *To the best of our knowledge, this is the first regret result for adversarial RL with switching costs.* To prove Theorem 3, the main difficulty lies in capturing the effects of the arbitrarily-changing losses and multiple random visitations of each state-action pair in a super-episode. To overcome this difficulty, our new idea is to first upper-bound the loss regret based on the correlated loss feedback in a super-episode, and then relate these upper bounds across all super-episodes to the final regret. The first step relies on the proof of Lemma 1, and the second step relies on another lemma in Appendix D.1 that transfers the original regret formulation to a form based on the losses from the entire super-episode. Please see Appendix D for details and the proof of Theorem 3.

Further, in Theorem 4 below, we show that SEEDS attains a trade-off between the loss regret and switching costs that matches the trade-off in Theorem 2. The proof of Theorem 4 follows the loss-regret bound of SEEDS proved in Appendix D and the trivial switching-cost bound $\beta \cdot \lceil \frac{T}{\tau} \rceil$. Please see the end of Appendix D for details.

---

**Algorithm 2** SEEDS-Unknown Transition (SEEDS-UT)

---

**Parameters:** $\eta = \tilde{\Theta}\left(\beta^{-1/3}H^{1/3}(SA)^{-1/3}T^{-2/3}\right)$, $\tau = \tilde{\Theta}\left(\beta^{2/3}H^{-2/3}(SA)^{-1/3}T^{1/3}\right)$, $\gamma = \tilde{\Theta}\left(\beta^{1/3}H^{2/3}(SA)^{-2/3}T^{-1/2}\right)$, and $0 < \delta < 1$.

**Initialization:** $\hat{q}_{[1]}^{\text{SEEDS-UT},\mathcal{P}}(s',s,a) = \frac{1}{S_{h+1}S_h A}$ and $M_{[1]}(s',s,a) = N_{[1]}(s,a) = 0$, for all $(s',s,a) \in \mathcal{S}_{h+1} \times \mathcal{S}_h \times \mathcal{A}$ and all $h$. $\mathcal{P}_{[1]}$ contains all possible transition functions. Choose $\pi_{[1]}^{\text{SEEDS-UT}} = \pi^{\hat{q}_{[1]}^{\text{SEEDS-UT},\mathcal{P}}}$ according to (2) and (5).

**for** $u = 1 : \lceil \frac{T}{\tau} \rceil$ **do**
    **for** $t = (u-1)\tau + 1 : \min\{u\tau, T\}$ **do**
        *Step 1:* Execute the updated policy $\pi_{[u]}^{\text{SEEDS-UT}} = \pi^{\hat{q}_{[u]}^{\text{SEEDS-UT},\mathcal{P}}}$.
    **end for**
    At the end of super-episode $u$,
    *Step 2:* Estimate the losses $\hat{l}_{[u]}^{\text{SEEDS-UT}}(s,a)$ for all $(s,a)$ according to (12).
    *Step 3:* Estimate the transition-function set $\mathcal{P}_{[u+1]}$ according to (14).
    *Step 4:* Update the occupancy measure $\hat{q}_{[u+1]}^{\text{SEEDS-UT},\mathcal{P}}(s',s,a)$ according to (10), but subject to a different constraint $q \in \mathbb{C}\left(\mathcal{P}_{[u+1]}\right)$. Update the deterministic policy $\pi^{\hat{q}_{[u+1]}^{\text{SEEDS-UT},\mathcal{P}}}$ according to (2) and (5).
**end for**

---

**Theorem 4.** *Let* $\mathcal{N}^{\text{SEEDS}} \triangleq \lceil \frac{T}{\tau} \rceil$. *Then, with the switching costs equal to* $O\left(\beta \cdot \mathcal{N}^{\text{SEEDS}}\right)$, *SEEDS can achieve a loss regret upper-bounded by* $\tilde{O}\left(\sqrt{\frac{HSA}{\mathcal{N}^{\text{SEEDS}}}} \cdot T\right)$.

## 6   THE CASE WHEN THE TRANSITION FUNCTION IS UNKNOWN

In this section, we study a more challenging case when the transition function is *unknown*. We propose a novel algorithm (please see Algorithm 2) with a regret that matches the lower bound in (7) in terms of the dependency on all parameters, except with a small factor of $\tilde{O}(H^{1/3})$. Specifically, to address the new difficulty due to the *unknown* transition function $P$ in this case, we advance SEEDS into SEEDS-UT (where UT stands for "unknown transition") with three new components as we explain below.

1. Since the transition function $P$ is unknown, updating the occupancy measure $\hat{q}(s,a)$ (as in SEEDS) is not good enough. Instead, SEEDS-UT updates the occupancy measure $\hat{q}(s',s,a)$ to take state transitions into consideration.

2. Since the transition function $P$ is unknown, the updated occupancy measure could be different from the true one. To resolve this issue, we generalize the method in Neu (2015), with a difference to handle the random sequence of the state-action pairs visited in each super-episode. Specifically, SEEDS-UT estimates the loss for each super-episode $u$ as follows (*Step-2* in Algorithm 2),

$$\hat{l}_{[u]}^{\text{SEEDS-UT}}(s,a) = \sum_{j=1}^{J_{[u]}} \frac{l_{t_j(s,a)}(s,a)}{\mathcal{Q}_{[u]}^{\gamma}(s,a)} \mathbf{1}_{\{(s,a)\text{ was visited in episodes } t_1(s,a),\ldots,t_{J_{[u]}}(s,a) \text{ of super-episode } u\}}, \quad (12)$$

where $\mathcal{Q}_{[u]}^{\gamma}(s,a) \triangleq \max_{q \in \mathbb{C}(\mathcal{P}_{[u]})} q(s,a) + \gamma$ is the sum of the largest probability of visiting $(s,a)$ among all occupancy measures in $\mathbb{C}(\mathcal{P}_{[u]})$ and a tunable parameter $\gamma > 0$, and $\mathcal{P}_{[u]}$ is a transition-function set that we will introduce soon. Note that (12) is another application of our idea in (8) for estimating losses in a problem with state transitions and adversarial losses.

In Lemma 2 below, we show that the gap between the expectation of the estimated loss and the true loss is controlled by the parameter $\gamma$. The proof of Lemma 2 is provided in Appendix F. We use $\mathcal{F}_{[u]}$ to denote the $\sigma$-algebra generated by the observation of SEEDS-UT before super-episode $u$.

**Lemma 2.** *The conditional expectation of the estimated loss designed in (12) is equal to*

$$\mathbb{E}\left[\hat{l}_{[u]}^{\text{SEEDS-UT}}(s,a) \Big| \mathcal{F}_{[u]}\right] = \frac{q_{[u]}^{\text{SEEDS-UT},P}(s,a)}{\max_{q \in \mathbb{C}(\mathcal{P}_{[u]})} q(s,a)+\gamma} \cdot l_{[u]}(s,a), \text{ for all } (s,a), \quad (13)$$

*where the expectation is taken with respect to the randomness of the episodes $t_1(s,a), ..., t_{J_{[u]}}(s,a)$, in which $(s,a)$ was visited, $q_{[u]}^{SEEDS\text{-}UT,P}(s,a)$ is the true occupancy measure of SEEDS-UT conditioned on $\mathcal{F}_{[u]}$, and $l_{[u]}(s,a) = \sum_{t=(u-1)\tau+1}^{\min\{u\tau,T\}} l_t(s,a)$ is the true loss of $(s,a)$ in super-episode $u$.*

Lemma 2 shows that, as long as $\mathcal{P}_{[u]}$ is sufficiently good for estimating the true transition function $P$ (we will show how to construct such a $\mathcal{P}_{[u]}$ below), by carefully tuning $\gamma$, the bias caused by $\max_{q \in \mathbb{C}(\mathcal{P}_{[u]})} q(s,a) + \gamma$ (i.e., $\mathcal{Q}_{[u]}^{\gamma}(s,a)$) should be sufficiently small, so that the estimated loss is still sufficiently accurate.

3. Since the transition function $P$ is unknown, the constraint in (10) is no longer known. To resolve this issue, we generalize the method in Jin et al. (2020a), with a difference to handle the samples from the whole super-episode. Specifically, at the end of each super-episode, SEEDS-UT collects the samples from the whole super-episode to update the empirical transition probability $\bar{P}_{[u+1]}(s'|s,a) = \frac{M_{[u+1]}(s',s,a)}{\max\{N_{[u+1]}(s,a),1\}}$, where $M_{[u+1]}(s',s,a)$ and $N_{[u+1]}(s,a)$ denote the number of times visiting $(s',s,a)$ and $(s,a)$ before super-episode $u+1$, respectively. Then, based on the empirical Bernstein bound (Maurer & Pontil, 2009), SEEDS-UT constructs a transition-function set $\mathcal{P}$ as follows (*Step-3* in Algorithm 2),

$$\mathcal{P}_{[u+1]} = \left\{ \hat{P}_{[u+1]} : \left| \hat{P}_{[u+1]}(s'|s,a) - \bar{P}_{[u+1]}(s'|s,a) \right| \leq \epsilon_{[u+1]}(s',s,a), \text{ for all } (s',s,a) \right\}, \quad (14)$$

where $\epsilon_{[u+1]}(s',s,a) = 2\sqrt{\frac{\bar{P}_{[u+1]}(s',s,a) \ln \frac{TSA}{\delta}}{\max\{N_{[u+1]}(s,a)-1,1\}}} + \frac{14 \ln \frac{TSA}{\delta}}{3\max\{N_{[u+1]}(s,a)-1,1\}}$, and $\delta \in (0,1)$ is the confidence parameter. Finally, the occupancy measure $\hat{q}_{[u+1]}^{SEEDS\text{-}UT,\mathcal{P}}(s',s,a)$ is updated according to (10), but subject to a different constraint $q \in \mathbb{C}\left(\mathcal{P}_{[u+1]}\right)$ (*Step-4* in Algorithm 2).

We characterize the regret of SEEDS-UT in Theorem 5 below.

**Theorem 5.** *Consider adversarial RL with switching costs introduced in Sec. 3. When the transition function $P$ is unknown, with probability $1-\delta$, the regret of SEEDS-UT is upper-bounded as follows,*

$$R^{SEEDS\text{-}UT}(T) \leq \tilde{O}\left( \beta^{1/3} H^{2/3} (SA)^{1/3} T^{2/3} \left( \ln \frac{TSA}{\delta} \right)^{1/2} \right). \quad (15)$$

Theorem 5 shows that the regret of SEEDS-UT matches the lower bound in (7) in terms of the dependency on $T$, $S$, $A$, and $\beta$, except with a small factor of $\tilde{O}(H^{1/3})$. That is, the regret of SEEDS-UT is near-optimal. *To the best of our knowledge, this is the first regret result for adversarial RL with switching cost when the transition function is unknown.* To prove Theorem 5, the main difficulty is that, due to the delayed switching and unknown transition function, the losses of SEEDS-UT in the episodes of any super-episode are correlated and the true occupancy measure is unknown. As a result, the existing analytical ideas in adversarial RL without switching costs and adversarial bandit learning with switching costs do not work here. To overcome these new difficulties, our analysis involves several new ideas, e.g., we construct a series in (35) to handle multiple random visitations of each state-action pairs, and we establish a super-episodic version of concentration in *Step-2-iii* of Appendix G by relating the second-order moment of the estimated loss that we design to the true loss and the length $\tau$ of a super-episode. Please see Appendix G for the detailed proof of Theorem 5.

## 7 CONCLUSION AND FUTURE WORK

In this paper, we make the first effort towards addressing the challenge of switching costs in adversarial RL. First, we provide a lower bound that shows that the best achieved regret in static RL with switching costs (as well as adversarial RL without switching costs) is no longer achievable. In addition, we characterize precisely the new trade-off between the loss regret and switching costs, which shows that the adversarial nature of RL necessarily requires more switches to achieve a low loss regret. Moreover, we propose two novel switching-reduced algorithms with regrets that match our lower bound when the transition function is known, and match our lower bound within a small factor of $\tilde{O}(H^{1/3})$ when the transition function is unknown.

Several future directions are worth pursuing. First, it is important to study adversarial RL with switching costs in linear and more general MDP settings. Another interesting future work is to extend our study to the dynamic regret, which allows the optimal policy to change over time.

ACKNOWLEDGMENTS

The work of M. Shi has been partly supported by NSF grant NSF AI Institute (AI-EDGE) CNS-2112471. The work of Y. Liang has been partly supported by NSF grants NSF AI-EDGE CNS-2112471 and RINGS-2148253. The work of N. Shroff has been partly supported by NSF grants NSF AI-EDGE CNS-2112471, CNS-2106933, 2007231, CNS-1955535, and CNS-1901057, and in part by Army Research Office under Grant W911NF-21-1-0244.

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

## A    PROOF OF THEOREM 1

Note that the bandit setting is a special case (when $S = H = 1$) of our MDP setting. Thus, the lower bound for the adversarial bandit setting in Shi et al. (2022) serves as a lower bound in our MDP setting. However, the direct use of such a lower bound from bandits will not be good enough for the MDP case that we study in this paper. To get the lower bound in Theorem 1, the most challenging and interesting part is to design the lower-bound instance. Notice that a lower-bound transition is constructed for stochastic MDP in Qiao et al. (2022), which shows that the MDP setting is at least as difficult as multi-armed bandits with $\Omega(HSA)$ arms, and then a similar lower bound can be obtained based on the lower bound from bandits. In this section, we construct a new lower-bound instance. Specifically, we divide the state space $\mathcal{S}$ and construct special state transitions, such that the episodic reinforcement learning is reduced to $\Theta(S/H)$ chains of bandit learning. Notice that the lower-bound analysis in Shi et al. (2022) implies that, with the loss function $l_t$ upper-bounded by $H$, $A$ arms and $T$ time-slots, the regret of any bandit-learning algorithm with switching costs is at least $\tilde{\Omega}\left(\beta^{1/3}A^{1/3}(HT)^{2/3}\right)$ when $T \geq \max\{6H^2A, \beta\}$. Hence, the total regret from all $\Theta(S/H)$ chains of bandit learning is at least $\tilde{\Omega}\left(\beta^{1/3}A^{1/3}(H\frac{T}{S/H})^{2/3}\right) \cdot \Theta(S/H) = \tilde{\Omega}\left(\beta^{1/3}(HSA)^{1/3}T^{2/3}\right)$. Please see our detailed proof below.

*Proof.* **Lower-bound instance:** We consider a special instance where $S - 2$ is divisible by $H - 1$. First, we assign the states in the state space $\mathcal{S}$ to each layer as follows. The first layer contains a single sate, i.e., $\mathcal{S}_0 = \{s_0\}$. All episodes end with state $\mathcal{S}_H = \{s_H\}$. Moreover, the rest of the $S - 2$ states are assigned to each layer $h \in [1, H-1]$ evenly. That it, each layer $h \in [1, H-1]$ contains $\frac{S-2}{H-1}$ states. Following the sequence of the states at each layer, we call the index $i$ of the $i$-th state the "order" of it. In addition, the order $i$ of the states at layer $h$ of any episode is the same, e.g., the first state at layer $h$ is always the first state at layer $h$ for all episodes, and the second state at layer $h$ is always the second state at layer $h$ for all episodes. Moreover, all actions are available at each state $s \in \mathcal{S}$. Finally, based on this construction of the states and actions, we run independently the lower-bound algorithm for adversarial bandit learning with switching costs in (Shi et al., 2022) as a subroutine through all $i$-th states, for all $i = 1, ..., \frac{S-2}{H-1}$. That is, for each layer $h = 1, ..., H-1$, $P_h(s_i|s_i, a) = 1$ for all $a$, and $P_h(s_j|s_i, a) = 0$ for all $j \neq i$ and all $a$.

**Lower-bound analysis:** The lower-bound analysis in Shi et al. (2022) implies that, with the loss function $l_t$ upper-bounded by $H$, $A$ arms and $T$ time-slots, the regret (including both the loss regret and switching costs) of any bandit-learning algorithm with switching costs is at least $\tilde{\Omega}\left(\beta^{1/3}A^{1/3}(HT)^{2/3}\right)$. Notice that based on our lower-bound instance constructed above, there are $\frac{S-2}{H-1}$ chains of bandit learning. Hence, the total regret of any RL algorithm $\pi$ from all these $\frac{S-2}{H-1}$ chains of bandit learning can be lower-bounded as follows,

$$R^\pi(T) \geq \tilde{\Omega}\left(\beta^{1/3}A^{1/3}\left(H\frac{T}{\frac{S-2}{H-1}}\right)^{2/3}\right) \cdot \frac{S-2}{H-1} = \tilde{\Omega}\left(\beta^{1/3}(HSA)^{1/3}T^{2/3}\right) \qquad (16)$$

$\square$

## B    PROOF OF THEOREM 2

The proof of Theorem 2 follows the lower bound proved in Appendix A, but by considering the loss regret and switching costs separately.

*Proof.* To prove Theorem 2, we use the lower-bound instance that we constructed above for proving Theorem 1 in Appendix A. First, the lower-bound analysis in Shi et al. (2022) implies that, for adversarial bandit learning with the loss function $l_t$ upper-bounded by $H$, $A$ arms and $T$ time-slots, when the total switching cost is equal to $O(\beta \cdot \mathcal{N}^{\text{swi}})$, the loss regret can be lower-bounded by $\tilde{\Omega}\left(\sqrt{\frac{A}{\mathcal{N}^{\text{swi}}}} \cdot HT\right)$. Notice that there are $\frac{S-2}{H-1}$ chains of bandit learning in the lower-bound instance that we constructed in Appendix A. Thus, with a total switching cost equal to $O(\beta \cdot \mathcal{N}^{\text{swi}}) \triangleq$

$O(\beta \cdot \sum_{i=1}^{\frac{S-2}{H-1}} \mathcal{N}_i^{\text{swi}})$, the loss regret of any RL algorithm $\pi$ against the lower-bound instance that we constructed above can be lower-bounded as follows,

$$R_{\text{loss}}^{\pi}(T) \geq \sum_{i=1}^{\frac{S-2}{H-1}} \tilde{\Omega}\left(\sqrt{\frac{A}{\mathcal{N}_i^{\text{swi}}}} \cdot H \frac{T}{\frac{S-2}{H-1}}\right) = \tilde{\Omega}\left(\sqrt{\frac{HSA}{\mathcal{N}^{\text{swi}}}} \cdot T\right),$$

where the equality is because $\sum_{i=1}^{\frac{S-2}{H-1}} \sqrt{\frac{1}{\mathcal{N}_i^{\text{swi}}}} \geq \sqrt{\frac{1}{\mathcal{N}^{\text{swi}}}} \left(\frac{S-2}{H-1}\right)^{3/2}$. Finally, the second half part of Theorem 2 is trivially true, since it is the converse-negative proposition of the first half part that we have proved above.

$\square$

## C  PROOF OF LEMMA 1

*Proof.* First, since the expectation is taken with respect to the randomness of the episodes $t_1(s,a)$, ..., $t_{J_{[u]}}(s,a)$, in which the state-action pair $(s,a)$ was visited, the left-hand-side of (9) is equal to

$$\mathbb{E}\left[\hat{l}_{[u]}^{\text{SEEDS}}(s,a)\Big|\mathcal{F}_{[u]}\right] = \sum_{\substack{\{t_1(s,a),...,t_{J_{[u]}}(s,a)\} \\ \subseteq[(u-1)\tau+1,u\tau]}} \hat{l}_{[u]}^{\text{SEEDS}}(s,a) \cdot Pr\left[\{t_1(s,a),...,t_{J_{[u]}}(s,a)\}\big|\mathcal{F}_{[u]}\right].$$

Next, according to the definition of the estimated loss that we design in (8), we have

$$\mathbb{E}\left[\hat{l}_{[u]}^{\text{SEEDS}}(s,a)\Big|\mathcal{F}_{[u]}\right] = \sum_{\substack{\{t_1(s,a),...,t_{J_{[u]}}(s,a)\} \\ \subseteq[(u-1)\tau+1,u\tau]}} \sum_{j=1}^{J_{[u]}} \frac{l_{t_j(s,a)}(s,a)}{\hat{q}_{[u]}^{\text{SEEDS},P}(s,a)}$$

$$\cdot \mathbf{1}_{\{(s,a) \text{ was visited in episodes } t_1(s,a),...,t_{J_{[u]}}(s,a) \text{ of super-episode } u\}} \cdot Pr\left[\{t_1(s,a),...,t_{J_{[u]}}(s,a)\}\big|\mathcal{F}_{[u]}\right].$$

In the following, we prove that

$$\sum_{\substack{\{t_1(s,a),...,t_{J_{[u]}}(s,a)\} \\ \subseteq[(u-1)\tau+1,u\tau]}} \sum_{j=1}^{J_{[u]}} \frac{l_{t_j(s,a)}(s,a)}{\hat{q}_{[u]}^{\text{SEEDS},P}(s,a)} \cdot \mathbf{1}_{\{(s,a) \text{ was visited in episodes } t_1(s,a),...,t_{J_{[u]}}(s,a) \text{ of super-episode } u\}}$$

$$\cdot Pr\left[\{t_1(s,a),...,t_{J_{[u]}}(s,a)\}\big|\mathcal{F}_{[u]}\right] = \sum_{t=(u-1)\tau+1}^{u\tau} \hat{q}_{[u]}^{\text{SEEDS},P}(s,a) \cdot \frac{l_t(s,a)}{\hat{q}_{[u]}^{\text{SEEDS},P}(s,a)}.$$

That is, under our new design of the estimated loss in (8), summing over all possible sets of the random episodes where the state-action pair was visited (i.e., the outer sum on the left-hand-side) is equivalent to summing over all deterministic episodes from the beginning to the end of a super-episode (i.e., the sum on the right-hand-side).

This is because first, relying on the above indicator function on the left-hand-side, the sum of the total *observed* loss in a super-episode over all possible sets $\{t_1(s,a),...,t_{J_{[u]}}(s,a)\}$ is equivalent to the sum of the total *true* loss in each episode of a super-episode based on whether the episode is observed. Therefore, we have

$$\mathbb{E}\left[\hat{l}_{[u]}^{\text{SEEDS}}(s,a)\Big|\mathcal{F}_{[u]}\right] = \sum_{t=(u-1)\tau+1}^{u\tau} \sum_{\substack{\{t_1(s,a),...,t_{J_{[u]}}(s,a)\}: \\ t\in\{t_1(s,a),...,t_{J_{[u]}}(s,a)\}}} \frac{l_t(s,a)}{\hat{q}_{[u]}^{\text{SEEDS},P}(s,a)}$$

$$\cdot Pr\left[\{t_1(s,a),...,t_{J_{[u]}}(s,a)\}\big|\mathcal{F}_{[u]}\right]. \qquad (17)$$

In addition, since the transition function $P$ is known, conditioned on $\mathcal{F}_{[u]}$, the probability of visiting each state-action pair $(s,a)$ in an episode $t$ of super-episode $u$ is equal to the occupancy measure

$\hat{q}_{[u]}^{\text{SEEDS},P}(s, a)$, i.e.,

$$\sum_{\substack{\{t_1(s,a),...,t_{J_{[u]}}(s,a)\}: \\ t \in \{t_1(s,a),...,t_{J_{[u]}}(s,a)\}}} Pr\left[\{t_1(s,a),...,t_{J_{[u]}}(s,a)\}\big|\mathcal{F}_{[u]}\right] = \hat{q}_{[u]}^{\text{SEEDS},P}(s, a). \tag{18}$$

Finally, by combining (17) and (18), we have

$$\mathbb{E}\left[\hat{l}_{[u]}^{\text{SEEDS}}(s, a)\Big|\mathcal{F}_{[u]}\right] = \sum_{t=(u-1)\tau+1}^{u\tau} \hat{q}_{[u]}^{\text{SEEDS},P}(s, a) \cdot \frac{l_t(s, a)}{\hat{q}_{[u]}^{\text{SEEDS},P}(s, a)}$$

$$= \sum_{t=(u-1)\tau+1}^{u\tau} l_t(s, a) = l_{[u]}(s, a).$$

$\square$

## D    PROOF OF THEOREM 3 AND THEOREM 4

Since the total switching cost of SEEDS is trivially upper-bounded by $\beta \cdot \lceil \frac{T}{\tau} \rceil$, to prove Theorem 3, we focus on upper-bounding the loss regret of SEEDS, i.e.,

$$R_{\text{loss}}^{\text{SEEDS}}(T) = \max_{q \in \mathbb{C}(P)} \mathbb{E}\left[\sum_{t=1}^{T} \left\langle q_t^{\text{SEEDS},P} - q, l_t \right\rangle \Bigg| \text{SEEDS}, P\right]$$

$$\triangleq \mathbb{E}\left[\sum_{t=1}^{T} \left\langle q_t^{\text{SEEDS},P} - q^{\pi^*}, l_t \right\rangle \Bigg| \text{SEEDS}, P\right].$$

To upper-bound the loss regret, the main difficulty lies in capturing the effects of the arbitrarily-changing losses and multiple random visitations of each state-action pair in a super-episode. To overcome this difficulty, our proof of Theorem 3 first upper-bounds the loss regret based on the correlated loss feedback in a super-episode (which relies on our new design of the estimated loss in (8) and Lemma 1), and then relates these upper bounds across all super-episodes to the final regret (which relies on another lemma, Lemma 3 below, which transfers the original regret formulation to a form based on the losses from the entire super-episode).

Specifically, for each super-episode, we first relate the true occupancy measure $q_t^{\text{SEEDS},P}$ to the un-constrained solution $\tilde{q}_{[u+1]}^{\text{SEEDS},P}$ to (10). Then, we relate $\tilde{q}_{[u+1]}^{\text{SEEDS},P}$ to the optimal offline occupancy measure $q^{\pi^*}$. The gaps between them are upper-bound mainly by using Lemma 1. Finally, by combining all the loss gaps (according to Lemma 3 and super-episodic version of online mirror descent) and the switching-cost upper-bound $\beta \lceil \frac{T}{\tau} \rceil$, and tuning the parameters $\eta$ and $\tau$ as in Algorithm 1, we can get the regret of SEEDS in Theorem 3 and the trade-off in Theorem 4. Please see the detailed proofs of Theorem 3 and Theorem 4 in the next two subsections.

### D.1    PROOF OF THEOREM 3

*Proof.* **Step-1 (Bounding the switching costs):** Since SEEDS switches at most once in each super-episode, the total switching cost of SEEDS is upper-bounded by $\beta \cdot \lceil \frac{T}{\tau} \rceil$. In the following, we focus on upper-bounding the loss regret $R_{\text{loss}}^{\text{SEEDS}}(T)$.

**Step-2 (Bounding the loss regret):** First, since SEEDS applies the same occupancy measure for all episodes $t$ of the same super-episode $u$ and the transition function $P$ is known, conditioned on the history before super-episode $u$, the true occupancy measures of these episodes are the same. Then, according to Lemma 3 below, we can transfer the original regret formulation to a form based on the losses from the entire super-episode.

**Lemma 3.** *The loss regret $R_{loss}^{SEEDS}(T)$ of SEEDS is equal to*

$$\mathbb{E}\left[\sum_{t=1}^{T} \left\langle q_t^{SEEDS,P} - q^{\pi^*}, l_t \right\rangle \Bigg| SEEDS, P\right] = \mathbb{E}\left[\sum_{u=1}^{\mathcal{U}} \left\langle q_{[u]}^{SEEDS,P} - q^{\pi^*}, l_{[u]} \right\rangle \Bigg| SEEDS, P\right]. \tag{19}$$

Note that the occupancy measure and loss on the left-hand-side of (19) are for each episode $t$, while those on the right-hand-side of (19) are for each super-episode $u$. Please see Appendix E for the proof of Lemma 3.

Next, we use $\tilde{q}_{[u+1]}^{\text{SEEDS},P}$ to denote the unconstrained solution to (10), i.e.,

$$\tilde{q}_{[u+1]}^{\text{SEEDS},P} \triangleq \arg\min_{q} \left\{ \eta \cdot \left\langle q, \hat{l}_{[u]}^{\text{SEEDS}} \right\rangle + D_{\text{KL}}\left( q \,\middle\|\, \hat{q}_{[u]}^{\text{SEEDS},P} \right) \right\}.$$

Notice that $\hat{q}_{[u+1]}^{\text{SEEDS},P}$ is the constrained solution to (10), where the constraint is $q \in \mathbb{C}(P)$. It is not hard to get that

$$\tilde{q}_{[u+1]}^{\text{SEEDS},P}(s,a) = \hat{q}_{[u]}^{\text{SEEDS},P}(s,a) \cdot e^{-\eta \hat{l}_{[u]}^{\text{SEEDS}}(s,a)}. \tag{20}$$

To get (20), let us consider the function $f(q) = \eta \cdot \left\langle q, \hat{l}_{[u]}^{\text{SEEDS}} \right\rangle + D_{\text{KL}}\left( q \,\middle\|\, \hat{q}_{[u]}^{\text{SEEDS},P} \right)$. According to the definition of $D_{\text{KL}}(q\|q')$ right after (10), the derivative of function $f(q)$ is

$$\frac{\partial f(q)}{\partial q(s,a)} = \eta \cdot \hat{l}_{[u]}^{\text{SEEDS}}(s,a) + \ln \frac{q(s,a)}{\hat{q}_{[u]}^{\text{SEEDS},P}(s,a)}.$$

By letting the derivative to be $0$ and rearranging the terms, we have (20).

**Remark 1.** *Notice that, similar to the above steps that use standard convex optimization method to get (20), we can get the the final solution to (10) as follows,*

$$\hat{q}_{[u+1]}^{SEEDS,P}(s,a) = \frac{\hat{q}_{[u]}^{SEEDS,P}(s,a)e^{\delta(s,a|\hat{v}_{[u]},\hat{l}_{[u]})}}{z_{[u]}(\hat{v}_{[u]},h(s))},$$

*where* $z_{[u]}(v,h) = \sum\limits_{s\in\mathcal{S}_h, a\in\mathcal{A}} q_{[u]}(s,a)e^{\delta(s,a|v,\hat{l}_{[u]})}$, $\delta(s,a|v,l) = -\eta l(s,a) - \sum\limits_{s'\in\mathcal{S}} v(s')P(s'|s,a) + v(s)$, *and* $\hat{v}_{[u]} = \arg\min\limits_{v} \sum\limits_{h=0}^{H} \ln z_{[u]}(v,h)$. *This is consistent with the expression provided in Proposition 1 in Zimin & Neu (2013).*

Then, because of Lemma 3 and the fact that the calculated occupancy measure $\hat{q}_{[u]}^{\text{SEEDS},P}$ is equal to the true occupancy measure $q_{[u]}^{\text{SEEDS},P}$, we have

$$\mathbb{E}\left[ \sum_{t=1}^{T} \left\langle q_t^{\text{SEEDS},P} - q^{\pi^*}, l_t \right\rangle \,\middle|\, \text{SEEDS}, P \right] = \mathbb{E}\left[ \sum_{u=1}^{\mathcal{U}} \left\langle \hat{q}_{[u]}^{\text{SEEDS},P} - q^{\pi^*}, l_{[u]} \right\rangle \,\middle|\, \text{SEEDS}, P \right].$$

According to the linearity of expectation, we can decompose the loss regret into two terms that are easier to be bounded as follows,

$$\mathbb{E}\left[ \sum_{t=1}^{T} \left\langle q_t^{\text{SEEDS},P} - q^{\pi^*}, l_t \right\rangle \,\middle|\, \text{SEEDS}, P \right] = \sum_{u=1}^{\mathcal{U}} \mathbb{E}\left[ \left\langle \hat{q}_{[u]}^{\text{SEEDS},P} - q^{\pi^*}, l_{[u]} \right\rangle \,\middle|\, \text{SEEDS}, P \right]$$

$$= \sum_{u=1}^{\mathcal{U}} \mathbb{E}_{\mathcal{F}_{[u]}}\left[ \mathbb{E}\left[ \left\langle \hat{q}_{[u]}^{\text{SEEDS},P} - q^{\pi^*}, l_{[u]} \right\rangle \,\middle|\, \mathcal{F}_{[u]}, P \right] \right]$$

$$= \sum_{u=1}^{\mathcal{U}} \mathbb{E}_{\mathcal{F}_{[u]}}\left[ \mathbb{E}\left[ \left\langle \hat{q}_{[u]}^{\text{SEEDS},P} - \tilde{q}_{[u+1]}^{\text{SEEDS},P}, l_{[u]} \right\rangle \,\middle|\, \mathcal{F}_{[u]}, P \right] \right]$$

$$+ \sum_{u=1}^{\mathcal{U}} \mathbb{E}_{\mathcal{F}_{[u]}}\left[ \mathbb{E}\left[ \left\langle \tilde{q}_{[u+1]}^{\text{SEEDS},P} - q^{\pi^*}, l_{[u]} \right\rangle \,\middle|\, \mathcal{F}_{[u]}, P \right] \right], \tag{21}$$

where the second equality is because $\mathbb{E}[X] = \mathbb{E}[\mathbb{E}[X|Y]]$, the last equality is because of the linearity of the expectation, and we drop the condition on SEEDS since it is clear from the context.

Below, we focus on upper-bounding the two terms on the right-hand-side of (21) one-by-one.

*Step-2-i (Bounding the first term):* Since $e^x \geq 1 + x$, from (20) we have

$$\hat{q}_{[u]}^{\text{SEEDS},P}(s,a) - \tilde{q}_{[u+1]}^{\text{SEEDS},P}(s,a) \leq \eta \hat{q}_{[u]}^{\text{SEEDS},P}(s,a) \cdot \hat{l}_{[u]}^{\text{SEEDS}}(s,a).$$

Thus, the first term on the right-hand-side of (21) can be upper-bounded as follows,

$$\sum_{u=1}^{\mathcal{U}} \mathbb{E}_{\mathcal{F}_{[u]}} \left[ \mathbb{E} \left[ \left\langle \hat{q}_{[u]}^{\text{SEEDS},P} - \tilde{q}_{[u+1]}^{\text{SEEDS},P}, l_{[u]} \right\rangle \Big| \mathcal{F}_{[u]}, P \right] \right]$$
$$\leq \sum_{u=1}^{\mathcal{U}} \mathbb{E}_{\mathcal{F}_{[u]}} \left[ \mathbb{E} \left[ \sum_{s \in \mathcal{S}, a \in \mathcal{A}} \eta \hat{q}_{[u]}^{\text{SEEDS},P}(s,a) \cdot \hat{l}_{[u]}^{\text{SEEDS}}(s,a) \cdot l_{[u]}(s,a) \Big| \mathcal{F}_{[u]}, P \right] \right].$$

Then, according to the definition of the estimated loss that we design in (8), we have

$$\sum_{u=1}^{\mathcal{U}} \mathbb{E}_{\mathcal{F}_{[u]}} \left[ \mathbb{E} \left[ \left\langle \hat{q}_{[u]}^{\text{SEEDS},P} - \tilde{q}_{[u+1]}^{\text{SEEDS},P}, l_{[u]} \right\rangle \Big| \mathcal{F}_{[u]}, P \right] \right]$$
$$\leq \sum_{u=1}^{\mathcal{U}} \mathbb{E}_{\mathcal{F}_{[u]}} \left[ \mathbb{E} \left[ \sum_{s \in \mathcal{S}, a \in \mathcal{A}} \eta \hat{q}_{[u]}^{\text{SEEDS},P}(s,a) \sum_{j=1}^{J_{[u]}} \frac{l_{t_j(s,a)}(s,a)}{\hat{q}_{[u]}^{\text{SEEDS},P}(s,a)} \right. \right.$$
$$\left. \left. \cdot \mathbf{1}_{\{(s,a) \text{ was visited in episodes } t_1(s,a),\ldots,t_{J_{[u]}}(s,a) \text{ of super-episode } u\}} \cdot l_{[u]}(s,a) \Big| \mathcal{F}_{[u]}, P \right] \right]$$
$$\leq \sum_{u=1}^{\mathcal{U}} \mathbb{E}_{\mathcal{F}_{[u]}} \left[ \mathbb{E} \left[ \sum_{s \in \mathcal{S}, a \in \mathcal{A}} \eta \left( l_{[u]}(s,a) \right)^2 \Big| \mathcal{F}_{[u]}, P \right] \right]$$
$$\leq \eta SA \left\lceil \frac{T}{\tau} \right\rceil \tau^2, \tag{22}$$

where the second inequality is because $\sum_{j=1}^{J_{[u]}} l_{t_j(s,a)}(s,a) \leq \sum_{t=(u-1)\tau+1}^{\min\{u\tau, T\}} l_t(s,a) = l_{[u]}(s,a)$, and the last inequality is because $l_{[u]}(s,a) \leq \tau$ and $\mathcal{U} = \left\lceil \frac{T}{\tau} \right\rceil$.

*Step-2-ii (Bounding the second term):* According to online mirror descent (Rakhlin et al., 2009; Zimin & Neu, 2013), we have the following inequality for the unconstrained solution $\tilde{q}_{[u+1]}^{\text{SEEDS},P}$ to (10),

$$\left\langle q - \tilde{q}_{[u+1]}^{\text{SEEDS},P}, \eta \cdot \hat{l}_{[u]}^{\text{SEEDS}} + \frac{\partial D_{\text{KL}}(q \| \hat{q}_{[u]}^{\text{SEEDS},P})}{\partial q} \Bigg|_{q = \tilde{q}_{[u+1]}^{\text{SEEDS},P}} \right\rangle \geq 0, \text{ for all } q.$$

Since $\dfrac{\partial D_{\text{KL}}(q \| \hat{q}_{[u]}^{\text{SEEDS},P})}{\partial q} \Bigg|_{q = \tilde{q}_{[u+1]}^{\text{SEEDS},P}} = \ln \left( \dfrac{\tilde{q}_{[u+1]}^{\text{SEEDS},P}}{\hat{q}_{[u]}^{\text{SEEDS},P}} \right)$, by rearranging the terms, we have

$$\left\langle \tilde{q}_{[u+1]}^{\text{SEEDS},P} - q, \eta \cdot \hat{l}_{[u]}^{\text{SEEDS}} \right\rangle \leq \left\langle q - \tilde{q}_{[u+1]}^{\text{SEEDS},P}, \ln \left( \frac{\tilde{q}_{[u+1]}^{\text{SEEDS},P}}{\hat{q}_{[u]}^{\text{SEEDS},P}} \right) \right\rangle, \text{ for all } q.$$

By adding and subtracting terms on the right-hand-side, we have

$$\left\langle \tilde{q}_{[u+1]}^{\text{SEEDS},P} - q, \eta \cdot \hat{l}_{[u]}^{\text{SEEDS}} \right\rangle$$

$$\leq \left[ \sum_{s\in\mathcal{S},a\in\mathcal{A}} q(s,a)\ln\frac{q(s,a)}{\hat{q}_{[u]}^{\text{SEEDS},P}(s,a)} - \sum_{s\in\mathcal{S},a\in\mathcal{A}}\left[ q(s,a) - \hat{q}_{[u]}^{\text{SEEDS},P}(s,a)\right]\right]$$

$$- \left[ \sum_{s\in\mathcal{S},a\in\mathcal{A}} \tilde{q}_{[u+1]}^{\text{SEEDS},P}(s,a)\ln\frac{\tilde{q}_{[u+1]}^{\text{SEEDS},P}(s,a)}{\hat{q}_{[u]}^{\text{SEEDS},P}(s,a)} - \sum_{s\in\mathcal{S},a\in\mathcal{A}}\left[ \tilde{q}_{[u+1]}^{\text{SEEDS},P}(s,a) - \hat{q}_{[u]}^{\text{SEEDS},P}(s,a)\right]\right]$$

$$+ \left[ \sum_{s\in\mathcal{S},a\in\mathcal{A}}\left( q(s,a) - \hat{q}_{[u]}^{\text{SEEDS},P}(s,a)\right) + \sum_{s\in\mathcal{S},a\in\mathcal{A}} q(s,a)\ln\frac{\tilde{q}_{[u+1]}^{\text{SEEDS},P}(s,a)}{q(s,a)}\right.$$

$$\left. - \sum_{s\in\mathcal{S},a\in\mathcal{A}}\left( \tilde{q}_{[u+1]}^{\text{SEEDS},P}(s,a) - \hat{q}_{[u]}^{\text{SEEDS},P}(s,a)\right)\right]$$

$$= D_{\text{KL}}\left( q\big\|\hat{q}_{[u]}^{\text{SEEDS},P}\right) - D_{\text{KL}}\left( \tilde{q}_{[u+1]}^{\text{SEEDS},P}\big\|\hat{q}_{[u]}^{\text{SEEDS},P}\right) - D_{\text{KL}}\left( q\big\|\tilde{q}_{[u+1]}^{\text{SEEDS},P}\right), \text{ for all } q.$$

Then, together with Lemma 1, we have

$$\sum_{u=1}^{\mathcal{U}} \mathbb{E}_{\mathcal{F}_{[u]}}\left[ \mathbb{E}\left[ \left\langle \tilde{q}_{[u+1]}^{\text{SEEDS},P} - q^{\pi^*}, l_{[u]} \right\rangle \Big| \mathcal{F}_{[u]}, P \right]\right]$$

$$= \sum_{u=1}^{\mathcal{U}} \mathbb{E}_{\mathcal{F}_{[u]}}\left[ \mathbb{E}\left[ \left\langle \tilde{q}_{[u+1]}^{\text{SEEDS},P} - q^{\pi^*}, \hat{l}_{[u]}^{\text{SEEDS}} \right\rangle \Big| \mathcal{F}_{[u]}, P \right]\right]$$

$$\leq \frac{1}{\eta} \cdot \sum_{u=1}^{\mathcal{U}} \mathbb{E}_{\mathcal{F}_{[u]}}\left[ \mathbb{E}\left[ D_{\text{KL}}\left( q\big\|\hat{q}_{[u]}^{\text{SEEDS},P}\right) - D_{\text{KL}}\left( \tilde{q}_{[u+1]}^{\text{SEEDS},P}\big\|\hat{q}_{[u]}^{\text{SEEDS},P}\right)\right.\right.$$

$$\left.\left. - D_{\text{KL}}\left( q\big\|\tilde{q}_{[u+1]}^{\text{SEEDS},P}\right)\Big| \mathcal{F}_{[u]}, P\right]\right].$$

Since the intermediate terms get cancelled and the relative entropy is always non-negative, the second term on the right-hand-side of (21) can be upper-bounded as follows,

$$\sum_{u=1}^{\mathcal{U}} \mathbb{E}_{\mathcal{F}_{[u]}}\left[ \mathbb{E}\left[ \left\langle \tilde{q}_{[u+1]}^{\text{SEEDS},P} - q^{\pi^*}, l_{[u]} \right\rangle \Big| \mathcal{F}_{[u]}, P\right]\right] \leq \frac{D_{\text{KL}}(q\|\hat{q}_{[1]}^{\text{SEEDS},P})}{\eta} \leq \frac{H}{\eta}\ln\frac{SA}{H}. \tag{23}$$

**Step-3 (Final step):** Finally, by combining (22), (23) and the switching-cost upper-bound $\beta \cdot \left\lceil \frac{T}{\tau}\right\rceil$, and tuning the parameters $\eta$ and $\tau$ as in Algorithm 1, we have that the regret of SEEDS is upper-bounded by $O\left( \beta^{1/3}(HSA)^{1/3} T^{2/3}\right)$.

$\square$

## D.2 PROOF OF THEOREM 4

By considering the loss-regret bound that we prove above and the switching-cost bound separately, we can prove Theorem 4.

*Proof.* According to (22) and (23) above, with the total switching cost equal to $O\left( \beta \cdot \left\lceil \frac{T}{\tau}\right\rceil\right) = O(\beta \cdot \mathcal{N}^{\text{SEEDS}})$, the loss regret of SEEDS is upper-bounded as follows,

$$R_{\text{loss}}^{\text{SEEDS}}(T) \leq \tilde{O}\left( \eta SAT\tau + \frac{H}{\eta}\right) = \tilde{O}\left( \sqrt{HSAT\tau}\right) = \tilde{O}\left( \sqrt{\frac{HSA}{\mathcal{N}^{\text{SEEDS}}}}\cdot T\right), \tag{24}$$

where the first equality is by tuning $\eta = \sqrt{\frac{H}{SAT\tau}}$, and the last equality is because $\mathcal{N}^{\text{SEEDS}} \triangleq \left\lceil \frac{T}{\tau}\right\rceil$.

$\square$

# E  PROOF OF LEMMA 3

For the convenience of the reader, we re-state Lemma 3 below.

**Lemma 3.** *The loss regret $R_{loss}^{SEEDS}(T)$ of SEEDS is equal to*

$$\mathbb{E}\left[\sum_{t=1}^{T}\left\langle q_t^{SEEDS,P}-q^{\pi^*},l_t\right\rangle\middle|SEEDS,P\right]=\mathbb{E}\left[\sum_{u=1}^{\mathcal{U}}\left\langle q_{[u]}^{SEEDS,P}-q^{\pi^*},l_{[u]}\right\rangle\middle|SEEDS,P\right]. \quad (25)$$

*Proof.* We drop the condition on SEEDS since it is clear from the context. First, according to the linearity of expectation, we have that the left-hand-side of (25) is equal to

$$\mathbb{E}\left[\sum_{t=1}^{T}\left\langle q_t^{SEEDS,P}-q^{\pi^*},l_t\right\rangle\middle|P\right]=\sum_{u=1}^{\mathcal{U}}\mathbb{E}\left[\sum_{t=(u-1)\tau+1}^{\min\{u\tau,T\}}\left\langle q_t^{SEEDS,P}-q^{\pi^*},l_t\right\rangle\middle|P\right].$$

Then, since conditioned on the history before super-episode $u$, the true occupancy measures for all episodes $t$ of the same super-episode $u$ are the same, we have

$$\mathbb{E}\left[\sum_{t=1}^{T}\left\langle q_t^{SEEDS,P}-q^{\pi^*},l_t\right\rangle\middle|P\right]=\sum_{u=1}^{\mathcal{U}}\mathbb{E}_{\mathcal{F}_{[u]}}\left[\mathbb{E}\left[\sum_{t=(u-1)\tau+1}^{\min\{u\tau,T\}}\left\langle q_t^{SEEDS,P}-q^{\pi^*},l_t\right\rangle\middle|\mathcal{F}_{[u]},P\right]\right]$$

$$=\sum_{u=1}^{\mathcal{U}}\mathbb{E}_{\mathcal{F}_{[u]}}\left[\mathbb{E}\left[\sum_{t=(u-1)\tau+1}^{\min\{u\tau,T\}}\left\langle q_{[u]}^{SEEDS,P}-q^{\pi^*},l_t\right\rangle\middle|\mathcal{F}_{[u]},P\right]\right].$$

Finally, since the true loss is $l_{[u]}(s,a)=\sum_{t=(u-1)\tau+1}^{\min\{u\tau,T\}}l_t$, we have

$$\mathbb{E}\left[\sum_{t=1}^{T}\left\langle q_t^{SEEDS,P}-q^{\pi^*},l_t\right\rangle\middle|P\right]=\sum_{u=1}^{\mathcal{U}}\mathbb{E}_{\mathcal{F}_{[u]}}\left[\mathbb{E}\left[\left\langle q_{[u]}^{SEEDS,P}-q^{\pi^*},l_{[u]}\right\rangle\middle|\mathcal{F}_{[u]},P\right]\right]$$

$$=\mathbb{E}\left[\sum_{u=1}^{\mathcal{U}}\left\langle q_{[u]}^{SEEDS,P}-q^{\pi^*},l_{[u]}\right\rangle\middle|P\right].$$

$\square$

# F  PROOF OF LEMMA 2

The proof is similar to the proof of Lemma 1 in Appendix C.

*Proof.* First, since the expectation is taken with respect to the randomness of the episodes $t_1(s,a)$, ..., $t_{J_{[u]}}(s,a)$, in which the state-action pair $(s,a)$ was visited, the left-hand-side of (13) is equal to

$$\mathbb{E}\left[\hat{l}_{[u]}^{SEEDS\text{-}UT}(s,a)\middle|\mathcal{F}_{[u]}\right]$$

$$=\sum_{\substack{\{t_1(s,a),...,t_{J_{[u]}}(s,a)\}\\\subseteq[(u-1)\tau+1,u\tau]}}\hat{l}_{[u]}^{SEEDS\text{-}UT}(s,a)\cdot Pr\left[\{t_1(s,a),...,t_{J_{[u]}}(s,a)\}\middle|\mathcal{F}_{[u]}\right].$$

Next, according to the definition of the estimated loss that we design in (12), we have

$$\mathbb{E}\left[\hat{l}_{[u]}^{SEEDS\text{-}UT}(s,a)\middle|\mathcal{F}_{[u]}\right]=\sum_{\substack{\{t_1(s,a),...,t_{J_{[u]}}(s,a)\}\\\subseteq[(u-1)\tau+1,u\tau]}}\sum_{j=1}^{J_{[u]}}\frac{l_{t_j(s,a)}(s,a)}{\mathcal{Q}_{[u]}^{\gamma}(s,a)}$$

$$\cdot \mathbf{1}_{\{(s,a)\text{ was visited in episodes }t_1(s,a),...,t_{J_{[u]}}(s,a)\text{ of super-episode }u\}}\cdot Pr\left[\{t_1(s,a),...,t_{J_{[u]}}(s,a)\}\middle|\mathcal{F}_{[u]}\right].$$

Then, relying on the above indicator function, the sum of the total *observed* loss in a super-episode over all possible sets $\{t_1(s,a), ..., t_{J_{[u]}}(s,a)\}$ is equivalent to the sum of the total *true* loss in each episode of a super-episode based on whether the episode is observed. Therefore, we have

$$\mathbb{E}\left[\hat{l}_{[u]}^{\text{SEEDS-UT}}(s,a)\Big|\mathcal{F}_{[u]}\right] = \sum_{t=(u-1)\tau+1}^{u\tau} \sum_{\substack{\{t_1(s,a),...,t_{J_{[u]}}(s,a)\}: \\ t\in\{t_1(s,a),...,t_{J_{[u]}}(s,a)\}}} \frac{l_t(s,a)}{\mathcal{Q}_{[u]}^{\gamma}(s,a)}$$
$$\cdot Pr\left[\{t_1(s,a),...,t_{J_{[u]}}(s,a)\}\Big|\mathcal{F}_{[u]}\right].$$

Finally, since conditioned on $\mathcal{F}_u$, the probability of visiting each state-action pair $(s,a)$ in an episode $t$ of super-episode $u$ is equal to the occupancy measure $q_{[u]}^{\text{SEEDS-UT},P}(s,a)$, we have

$$\mathbb{E}\left[\hat{l}_{[u]}^{\text{SEEDS-UT}}(s,a)\Big|\mathcal{F}_{[u]}\right] = \sum_{t=(u-1)\tau+1}^{u\tau} q_{[u]}^{\text{SEEDS-UT},P}(s,a) \cdot \frac{l_t(s,a)}{\mathcal{Q}_{[u]}^{\gamma}(s,a)}$$
$$= \frac{q_{[u]}^{\text{SEEDS-UT},P}(s,a)}{\mathcal{Q}_{[u]}^{\gamma}(s,a)} \sum_{t=(u-1)\tau+1}^{u\tau} l_t(s,a) = \frac{q_{[u]}^{\text{SEEDS-UT},P}(s,a)}{\mathcal{Q}_{[u]}^{\gamma}(s,a)} l_{[u]}(s,a).$$

$\square$

## G  PROOF OF THEOREM 5

Specifically, since the total switching cost of SEEDS-UT is trivially upper-bounded by $\beta \cdot \left\lceil \frac{T}{\tau} \right\rceil$, to prove Theorem 5, we focus on upper-bounding the loss regret $R_{\text{loss}}^{\text{SEEDS}}(T)$ of SEEDS-UT, i.e.,

$$R_{\text{loss}}^{\text{SEEDS-UT}}(T) = \max_{q\in\mathbb{C}(P)} \mathbb{E}\left[\sum_{t=1}^{T}\left\langle q_t^{\text{SEEDS-UT},P} - q, l_t\right\rangle \Big| \text{SEEDS-UT}, P\right]$$
$$\triangleq \mathbb{E}\left[\sum_{t=1}^{T}\left\langle q_t^{\text{SEEDS-UT},P} - q^{\pi^*}, l_t\right\rangle \Big| \text{SEEDS-UT}, P\right].$$

To upper-bound the loss regret, the main difficulties are that, due to the delayed switching and unknown transition function, the losses of SEEDS-UT in the episodes of any super-episode are correlated and the true occupancy measure is unknown. As a result, the existing analytical ideas in adversarial RL without switching costs (e.g., in Jin et al. (2020a)) and adversarial bandit learning with switching costs (e.g., in Shi et al. (2022)) do not apply here. To overcome these new difficulties, our proof of Theorem 5 involves several key new components. For example, since SEEDS-UT collects samples from a whole super-episode to estimate the transition-function set $\mathcal{P}$, each state-action pair could be visited multiple times and such visitations are random. As a result, the proof in Jin et al. (2020a), which requires each state-action pair to be visited at most once does not apply directly here. To resolve this difficulty, we construct a special series based on the collected samples to achieve an analyzable intermediate step for our proof of the final regret. Moreover, due to our new design of the estimated loss in (12), the concentration lemma for the loss based on the samples from only one episode in Jin et al. (2020a) does not apply. To resolve this difficulty, we establish a super-episodic version of concentration in our proof by bounding the second-order moment of the estimated loss.

Specifically, for each super-episode, we decompose the loss regret $\left\langle q_{[u]}^{\text{SEEDS-UT},P} - q^{\pi^*}, l_{[u]}\right\rangle$ into four parts that are easier to be upper-bounded as follows,

$$\left\langle q_{[u]}^{\text{SEEDS-UT},P} - q^{\pi^*}, l_{[u]}\right\rangle = \left\langle q_{[u]}^{\text{SEEDS-UT},P} - \hat{q}_{[u]}^{\text{SEEDS-UT},\mathcal{P}}, l_{[u]}\right\rangle + \left\langle \hat{q}_{[u]}^{\text{SEEDS-UT},\mathcal{P}}, l_{[u]} - \hat{l}_{[u]}^{\text{SEEDS-UT}}\right\rangle$$
$$+ \left\langle \hat{q}_{[u]}^{\text{SEEDS-UT},\mathcal{P}} - q^{\pi^*}, \hat{l}_{[u]}^{\text{SEEDS-UT}}\right\rangle + \left\langle q^{\pi^*}, \hat{l}_{[u]}^{\text{SEEDS-UT}} - l_{[u]}\right\rangle.$$

The first term on the right-hand-side is mainly the difference between the true occupancy measure and the updated occupancy measure. Intuitively, according to Bernstein inequality (Maurer & Pontil,

2009) and standard stochastic RL analysis, SEEDS-UT estimates the true transition function $P$ very well by using the transition-function set $\mathcal{P}$ in (14). Thus, based on the relation between the occupancy measure and the transition function in (4), SEEDS-UT should estimate the true occupancy measure very well. Hence, the first term should be upper-bounded and controllable. The second and fourth terms on the right-hand-side depends on the difference between the estimated loss and the true loss. According to Lemma 2, this gap should be controllable by tuning the parameter $\gamma$. The third term is similar to the loss regret in the case when the transition function is known. Thus, it can be upper-bounded similarly to our proof of Theorem 3 in Appendix D. Finally, by combining all these gaps and the switching-cost upper-bound $\beta \cdot \lceil \frac{T}{\tau} \rceil$, and tuning the parameters $\eta$, $\tau$ and $\gamma$ as in Algorithm 2, we get the regret of SEEDS-UT in Theorem 5. Please see the detailed proof below.

*Proof.* **Step-1 (Bounding the switching costs):** Since SEEDS-UT switches at most once in each super-episode, the total switching cost of SEEDS-UT is upper-bounded by $\beta \cdot \lceil \frac{T}{\tau} \rceil$. In the following, we focus on upper-bounding the loss regret $R_{\text{loss}}^{\text{SEEDS-UT}}(T)$.

**Step-2 (Bounding the loss regret):** We first show Lemma 4 below. Lemma 4 is critical for Lemma 3 to be true in this case with an unknown transition function.

**Lemma 4.** *For any two episodes $t_1$ and $t_2$, if the updated occupancy measures are the same, i.e., $\hat{q}_{t_1}(s', s, a) = \hat{q}_{t_2}(s', s, a)$ for any $(s', s, a)$, then the true occupancy measures are the same, i.e., $q_{t_1}(s, a) = q_{t_2}(s, a) = q_{[u]}(s, a)$ for any $(s, a)$, where $q_{[u]}(s, a)$ is the true occupancy measure for the super-episode $u$.*

The proof of Lemma 4 follows the conditions in (2)-(5). Since SEEDS-UT applies the same occupancy measure $\hat{q}_{[u]}^{\text{SEEDS-UT}, \mathcal{P}}$ for all episodes $t$ of the same super-episode $u$, according to Lemma 4, the true occupancy measure $q_t^{\text{SEEDS-UT}, P}$ of these episodes $t$ are the same. Thus, similar to the case with a known transition function, we can get an unknown-transition version of Lemma 3 here. Thus,

$$\mathbb{E}\left[\sum_{t=1}^{T}\left\langle q_t^{\text{SEEDS-UT}, P} - q^{\pi^*}, l_t\right\rangle \Big| P\right] = \mathbb{E}\left[\sum_{u=1}^{\mathcal{U}}\left\langle q_{[u]}^{\text{SEEDS-UT}, P} - q^{\pi^*}, l_{[u]}\right\rangle \Big| P\right].$$

We drop the condition on SEEDS-UT in the expectation here and in the following when it is clear from the context.

According to the linearity of expectation, we can decompose the loss regret into four terms that are easier to be bounded, i.e.,

$$\mathbb{E}\left[\sum_{t=1}^{T}\left\langle q_t^{\text{SEEDS-UT}, P} - q^{\pi^*}, l_t\right\rangle \Big| P\right] = \sum_{u=1}^{\mathcal{U}} \mathbb{E}\left[\left\langle q_{[u]}^{\text{SEEDS-UT}, P} - q^{\pi^*}, l_{[u]}\right\rangle \Big| P\right]$$

$$= \sum_{u=1}^{\mathcal{U}}\left\{\mathbb{E}_{\mathcal{F}_{[u]}}\left[\mathbb{E}\left[\left\langle q_{[u]}^{\text{SEEDS-UT}, P} - \hat{q}_{[u]}^{\text{SEEDS-UT}, \mathcal{P}}, l_{[u]}\right\rangle + \left\langle \hat{q}_{[u]}^{\text{SEEDS-UT}, \mathcal{P}}, l_{[u]} - \hat{l}_{[u]}^{\text{SEEDS-UT}}\right\rangle\right.\right.\right.$$

$$\left.\left.\left. + \left\langle \hat{q}_{[u]}^{\text{SEEDS-UT}, \mathcal{P}} - q^{\pi^*}, \hat{l}_{[u]}^{\text{SEEDS-UT}}\right\rangle + \left\langle q^{\pi^*}, \hat{l}_{[u]}^{\text{SEEDS-UT}} - l_{[u]}\right\rangle \Big| \mathcal{F}_{[u]}, P\right]\right]\right\}. \quad (26)$$

Below, we focus on upper-bounding the four terms on the right-hand-side of (26) one-by-one.

*Step-2-i (Bounding the first term):* Since $l_t(s, a) \leq 1$ for all state-action pairs $(s, a)$, we have $l_{[u]}(s, a) \leq \tau$ for all $(s, a)$. Thus, we have

$$\left\langle q_{[u]}^{\text{SEEDS-UT}, P} - \hat{q}_{[u]}^{\text{SEEDS-UT}, \mathcal{P}}, l_{[u]}\right\rangle \leq \tau \cdot \sum_{s \in \mathcal{S}, a \in \mathcal{A}} \left| q_{[u]}^{\text{SEEDS-UT}, P}(s, a) - \hat{q}_{[u]}^{\text{SEEDS-UT}, \mathcal{P}}(s, a)\right|.$$

The difference between the true occupancy measure and the updated occupancy measure on the right-hand-side depends on how good the transition-function set $\mathcal{P}$ in (14) is, and can be further upper-bounded by using Bernstein inequality (Maurer & Pontil, 2009). Below, we focus on bounding this difference. We use $\tilde{\pi}(a|s)$ to denote the probability of choosing action $a$ at state $s$. Specifically, first, according to the relation between the occupancy measure and the transition function in

(4), we have that for any state-action pair $(s_h, a_h) \in \mathcal{S}_h \times \mathcal{A}$ visited at stage $h$,

$$q^{\pi,P}(s_h, a_h) = \tilde{\pi}(a_h|s_h) \sum_{(s_i \in \mathcal{S}_i, a_i \in \mathcal{A})_{i=0}^{h-1}} \prod_{j=0}^{h-1} [\tilde{\pi}(a_j|s_j)P(s_{j+1}|s_j, a_j)],$$

where for simplicity, we drop the index $t$ for the states $s$ and actions $a$. Thus, the difference between the updated occupancy measure and the true occupancy measure can be upper-bounded as follows,

$$\left| \hat{q}_{[u]}^{\text{SEEDS-UT},\mathcal{P}}(s_h, a_h) - q_{[u]}^{\text{SEEDS-UT},P}(s_h, a_h) \right| = \tilde{\pi}_{[u]}^{\text{SEEDS-UT}}(a_h|s_h)$$

$$\cdot \sum_{(s_i \in \mathcal{S}_i, a_i \in \mathcal{A})_{i=0}^{h-1}} \prod_{j=0}^{h-1} \tilde{\pi}_{[u]}^{\text{SEEDS-UT}}(a_j|s_j) \left[ \prod_{j=0}^{h-1} \hat{P}_{[u]}(s_{j+1}|s_j, a_j) - \prod_{j=0}^{h-1} P(s_{j+1}|s_j, a_j) \right], \quad (27)$$

For the terms in the bracket $[\cdot]$, we have

$$\prod_{j=0}^{h-1} \hat{P}_{[u]}(s_{j+1}|s_j, a_j) - \prod_{j=0}^{h-1} P(s_{j+1}|s_j, a_j)$$

$$= \prod_{j=0}^{h-1} \hat{P}_{[u]}(s_{j+1}|s_j, a_j) - \prod_{j=0}^{h-1} P(s_{j+1}|s_j, a_j) \pm \sum_{k=1}^{h-1} \prod_{j=0}^{k-1} P(s_{j+1}|s_j, a_j) \prod_{j=k}^{h-1} \hat{P}_{[u]}(s_{j+1}|s_j, a_j)$$

$$= \sum_{k=0}^{h-1} \left[ \hat{P}_{[u]}(s_{k+1}|s_k, a_k) - P(s_{k+1}|s_k, a_k) \right] \prod_{j=0}^{k-1} P(s_{j+1}|s_j, a_j) \prod_{j=k}^{h-1} \hat{P}_{[u]}(s_{j+1}|s_j, a_j)$$

$$\leq \sum_{k=0}^{h-1} \tilde{\epsilon}_{[u]}(s_{k+1}|s_k, a_k) \prod_{j=0}^{k-1} P(s_{j+1}|s_j, a_j) \prod_{j=k}^{h-1} \hat{P}_{[u]}(s_{j+1}|s_j, a_j), \quad (28)$$

where

$$\tilde{\epsilon}_{[u]}(s_{k+1}|s_k, a_k) = O\left( \sqrt{\frac{P(s_{k+1}|s_k, a_k) \ln \frac{TSA}{\delta}}{\max\{N_{[u]}(s_k, a_k)\}, 1\}}} + \frac{\ln \frac{TSA}{\delta}}{\max\{N_{[u]}(s_k, a_k)\}, 1\}} \right) \quad (29)$$

shows how good SEEDS-UT estimates the true transition function, and the inequality is because of the empirical Bernstein inequality (Maurer & Pontil, 2009) and Lemma 8 in Jin et al. (2020a). Applying (27) and (28) to SEEDS-UT, we have

$$\left| \hat{q}_{[u]}^{\text{SEEDS-UT},\mathcal{P}}(s_h, a_h) - q_{[u]}^{\text{SEEDS-UT},P}(s_h, a_h) \right| \leq \sum_{k=0}^{h-1} \sum_{(s_i \in \mathcal{S}_i, a_i \in \mathcal{A})_{i=0}^{h-1}} \tilde{\epsilon}_{[u]}(s_{k+1}|s_k, a_k)$$

$$\cdot \left[ \tilde{\pi}_{[u]}^{\text{SEEDS-UT}}(a_k|s_k) \prod_{j=0}^{k-1} \tilde{\pi}_{[u]}^{\text{SEEDS-UT}}(a_j|s_j) P(s_{j+1}|s_j, a_j) \right]$$

$$\cdot \left[ \tilde{\pi}_{[u]}^{\text{SEEDS-UT}}(a_h|s_h) \prod_{j=k+1}^{h-1} \tilde{\pi}_{[u]}^{\text{SEEDS-UT}}(a_j|s_j) \hat{P}(s_{j+1}|s_j, a_j) \right]$$

$$= \sum_{k=0}^{h-1} \sum_{s_{k+1} \in \mathcal{S}_{k+1}, s_k \in \mathcal{S}_k, a_k \in \mathcal{A}} \tilde{\epsilon}_{[u]}(s_{k+1}|s_k, a_k) q_{[u]}^{\text{SEEDS-UT},P}(s_k, a_k) \hat{q}_{[u]}^{\text{SEEDS-UT},\mathcal{P}}(s_h, a_h|s_{k+1}).$$

$$(30)$$

Similarly, we can show that

$$\left| \hat{q}_{[u]}^{\text{SEEDS-UT},\mathcal{P}}(s_h, a_h|s_{k+1}) - q_{[u]}^{\text{SEEDS-UT},P}(s_h, a_h|s_{k+1}) \right|$$

$$= \sum_{j=k+1}^{h-1} \sum_{s_{j+1} \in \mathcal{S}_{j+1}, s_j \in \mathcal{S}_j, a_j \in \mathcal{A}} \tilde{\epsilon}_{[u]}(s_{j+1}|s_j, a_j) q_{[u]}^{\text{SEEDS-UT},P}(s_j, a_j|s_{k+1}) \hat{q}_{[u]}^{\text{SEEDS-UT},\mathcal{P}}(s_h, a_h|s_{j+1})$$

$$\leq \tilde{\pi}_{[u]}^{\text{SEEDS-UT}}(a_h|s_h) \sum_{j=k+1}^{h-1} \sum_{s_{j+1} \in \mathcal{S}_{j+1}, s_j \in \mathcal{S}_j, a_j \in \mathcal{A}} \tilde{\epsilon}_{[u]}(s_{j+1}|s_j, a_j) q_{[u]}^{\text{SEEDS-UT},P}(s_j, a_j|s_{k+1}). \quad (31)$$

Combining (30) and (31), we have

$$
\sum_{u=1}^{\mathcal{U}} \sum_{h=0}^{H-1} \sum_{(s_h,a_h)\in\mathcal{S}_h\times\mathcal{A}} \left| \hat{q}_{[u]}^{\text{SEEDS-UT},\mathcal{P}}(s_h,a_h) - q_{[u]}^{\text{SEEDS-UT},P}(s_h,a_h) \right|
$$

$$
\leq \sum_{u=1}^{\mathcal{U}} \sum_{h=0}^{H-1} \sum_{(s_h,a_h)\in\mathcal{S}_h\times\mathcal{A}} \sum_{k=0}^{h-1} \sum_{(s_{k+1},s_k,a_k)\in\mathcal{S}_{k+1}\times\mathcal{S}_k\times\mathcal{A}} \tilde{\epsilon}_{[u]}(s_{k+1}|s_k,a_k)q_{[u]}^{\text{SEEDS-UT},P}(s_k,a_k)
$$
$$
\cdot q_{[u]}^{\text{SEEDS-UT},P}(s_h,a_h|s_{k+1})
$$

$$
+ \sum_{u=1}^{\mathcal{U}} \sum_{h=0}^{H-1} \sum_{(s_h,a_h)\in\mathcal{S}_h\times\mathcal{A}} \sum_{k=0}^{h-1} \sum_{(s_{k+1},s_k,a_k)\in\mathcal{S}_{k+1}\times\mathcal{S}_k\times\mathcal{A}} \tilde{\epsilon}_{[u]}(s_{k+1}|s_k,a_k)q_{[u]}^{\text{SEEDS-UT},P}(s_k,a_k)
$$
$$
\cdot \left[ \tilde{\pi}_{[u]}^{\text{SEEDS-UT}}(a_h|s_h) \sum_{j=k+1}^{h-1} \sum_{(s_{j+1},s_j,a_j)\in\mathcal{S}_{j+1}\times\mathcal{S}_j\times\mathcal{A}} \tilde{\epsilon}_{[u]}(s_{j+1}|s_j,a_j)q_{[u]}^{\text{SEEDS-UT},P}(s_j,a_j|s_{k+1}) \right].
$$
$$
\tag{32}
$$

Since $\sum_{h=0}^{H-1} \sum_{(s_h,a_h)\in\mathcal{S}_h\times\mathcal{A}} q_{[u]}^{\text{SEEDS-UT},P}(s_h,a_h|s_{k+1}) = 1$ and $\sum_{h=0}^{H-1} \sum_{(s_h,a_h)\in\mathcal{S}_h\times\mathcal{A}} \tilde{\pi}_{[u]}^{\text{SEEDS-UT}}(a_h|s_h) \leq S$, from (32), we have

$$
\sum_{u=1}^{\mathcal{U}} \sum_{h=0}^{H-1} \sum_{(s_h,a_h)\in\mathcal{S}_h\times\mathcal{A}} \left| \hat{q}_{[u]}^{\text{SEEDS-UT},\mathcal{P}}(s_h,a_h) - q_{[u]}^{\text{SEEDS-UT},P}(s_h,a_h) \right|
$$

$$
\leq \sum_{u=1}^{\mathcal{U}} \sum_{k=0}^{H-1} \sum_{(s_{k+1},s_k,a_k)\in\mathcal{S}_{k+1}\times\mathcal{S}_k\times\mathcal{A}} \tilde{\epsilon}_{[u]}(s_{k+1}|s_k,a_k)q_{[u]}^{\text{SEEDS-UT},P}(s_k,a_k)
$$

$$
+ S \cdot \sum_{u=1}^{\mathcal{U}} \sum_{k=0}^{H-1} \sum_{j=k+1}^{H-1} \sum_{\substack{(s_{k+1},s_k,a_k)\in\mathcal{S}_{k+1}\times\mathcal{S}_k\times\mathcal{A} \\ (s_{j+1},s_j,a_j)\in\mathcal{S}_{j+1}\times\mathcal{S}_j\times\mathcal{A}}} \tilde{\epsilon}_{[u]}(s_{k+1}|s_k,a_k)q_{[u]}^{\text{SEEDS-UT},P}(s_k,a_k)
$$
$$
\cdot \tilde{\epsilon}_{[u]}(s_{j+1}|s_j,a_j)q_{[u]}^{\text{SEEDS-UT},P}(s_j,a_j|s_{k+1}). \tag{33}
$$

Let us focus on bounding the terms on the right-hand-side of (33) one-by-one. For the first term, we have

$$
\sum_{u=1}^{\mathcal{U}} \sum_{k=0}^{H-1} \sum_{(s_{k+1},s_k,a_k)\in\mathcal{S}_{k+1}\times\mathcal{S}_k\times\mathcal{A}} \tilde{\epsilon}_{[u]}(s_{k+1}|s_k,a_k)q_{[u]}^{\text{SEEDS-UT},P}(s_k,a_k)
$$

$$
= O\left( \sum_{u=1}^{\mathcal{U}} \sum_{k=0}^{H-1} \sum_{(s_{k+1},s_k,a_k)\in\mathcal{S}_{k+1}\times\mathcal{S}_k\times\mathcal{A}} q_{[u]}^{\text{SEEDS-UT},P}(s_k,a_k)\sqrt{\frac{P(s_{k+1}|s_k,a_k)\ln\frac{TSA}{\delta}}{\max\left\{N_{[u]}(s_k,a_k)\right\},1\}}} \right.
$$
$$
\left. + \frac{q_{[u]}^{\text{SEEDS-UT},P}(s_k,a_k)\ln\frac{TSA}{\delta}}{\max\left\{N_{[u]}(s_k,a_k)\right\},1\}} \right)
$$

$$
\leq O\left( \sum_{u=1}^{\mathcal{U}} \sum_{k=0}^{H-1} \sum_{(s_k,a_k)\in\mathcal{S}_k\times\mathcal{A}} q_{[u]}^{\text{SEEDS-UT},P}(s_k,a_k)\sqrt{\frac{S_{k+1}\ln\frac{TSA}{\delta}}{\max\left\{N_{[u]}(s_k,a_k)\right\},1\}}} \right.
$$
$$
\left. + \frac{q_{[u]}^{\text{SEEDS-UT},P}(s_k,a_k)\ln\frac{TSA}{\delta}}{\max\left\{N_{[u]}(s_k,a_k)\right\},1\}} \right),
$$

where the equality is according to the definition of $\tilde{\epsilon}_{[u]}(s_{k+1}|s_k,a_k)$ in (29), and the inequality is according to Cauchy-Schwarz inequality. Note that the difficulty to further bound the above terms is that each state-action pair could be visited multiple times in a super-episode $u$. To this end, we construct a series to achieve an analyzable intermediate step. Let us first imagine there is a

sequence of numbers based on the samples that are collected from each single episode. Then, we use $N_t(s_k, a_k)$ to denote the number of times visiting the state-action pair $(s_k, a_k)$ before episode $t$. Since $N_t(s_k, a_k)$ is non-decreasing as $t$ increases, i.e.,

$$N_{(u-1)\tau+1}(s_k, a_k) \le N_{(u-1)\tau+2}(s_k, a_k) \le ... \le N_{u\tau}(s_k, a_k) = N_{[u]}(s_k, a_k), \quad (34)$$

we have

$$\frac{q_{[u]}^{\text{SEEDS-UT},P}(s_k, a_k)}{\sqrt{\max\left\{N_{[u]}(s_k, a_k)\right\}, 1\}}} = \frac{q_{[u]}^{\text{SEEDS-UT},P}(s_k, a_k)}{\sqrt{\max\left\{N_{u\tau}(s_k, a_k)\right\}, 1\}}} \le ... \le \frac{q_{[u]}^{\text{SEEDS-UT},P}(s_k, a_k)}{\sqrt{\max\left\{N_{(u-1)\tau+1}(s_k, a_k)\right\}, 1\}}}.$$

Now, let us compare our regret bound before to a intermediate step that is based on this series, i.e.,

$$\sum_{u=1}^{\mathcal{U}} \sum_{k=0}^{H-1} \sum_{(s_{k+1}, s_k, a_k) \in \mathcal{S}_{k+1} \times \mathcal{S}_k \times \mathcal{A}} \tilde{\epsilon}_{[u]}(s_{k+1}|s_k, a_k) q_{[u]}^{\text{SEEDS-UT},P}(s_k, a_k)$$

$$\le O\left( \frac{1}{\tau} \sum_{u=1}^{\mathcal{U}} \sum_{k=0}^{H-1} \sum_{(s_k, a_k) \in \mathcal{S}_k \times \mathcal{A}} \sum_{t=(u-1)\tau+1}^{u\tau} q_{[u]}^{\text{SEEDS-UT},P}(s_k, a_k) \sqrt{\frac{S_{k+1} \ln \frac{TSA}{\delta}}{\max\left\{N_t(s_k, a_k)\right\}, 1\}}} \right.$$

$$\left. + \frac{q_{[u]}^{\text{SEEDS-UT},P}(s_k, a_k) \ln \frac{TSA}{\delta}}{\max\left\{N_t(s_k, a_k)\right\}, 1\}} \right)$$

$$\le O\left( \frac{1}{\tau} \sum_{k=0}^{H-1} \sqrt{S_k S_{k+1} A T \ln \frac{TSA}{\delta}} \right)$$

$$\le O\left( \frac{1}{\tau} H S \sqrt{A T \ln \frac{TSA}{\delta}} \right), \quad (35)$$

Let us now consider the second term on the right-hand-side of (33), which can be upper-bounded similarly to the steps above to bound the first term. First, according to the definition of $\tilde{\epsilon}_{[u]}(s_{k+1}|s_k, a_k)$ in (29), we have this second term is upper-bounded by

$$S \cdot O\left( \sum_{u=1}^{\mathcal{U}} \sum_{k=0}^{H-1} \sum_{j=k+1}^{H-1} \sum_{\substack{(s_{k+1}, s_k, a_k) \in \mathcal{S}_{k+1} \times \mathcal{S}_k \times \mathcal{A} \\ (s_{j+1}, s_j, a_j) \in \mathcal{S}_{j+1} \times \mathcal{S}_j \times \mathcal{A}}} \sqrt{\frac{P(s_{k+1}|s_k, a_k) \ln \frac{TSA}{\delta}}{\max\left\{N_{[u]}(s_k, a_k)\right\}, 1\}}} q_{[u]}^{\text{SEEDS-UT},P}(s_k, a_k) \right.$$

$$\cdot \sqrt{\frac{P(s_{j+1}|s_j, a_j) \ln \frac{TSA}{\delta}}{\max\left\{N_{[u]}(s_j, a_j)\right\}, 1\}}} q_{[u]}^{\text{SEEDS-UT},P}(s_j, a_j|s_{k+1}) + \ln \frac{TSA}{\delta}$$

$$\left. \cdot \sum_{u=1}^{\mathcal{U}} \sum_{k=0}^{H-1} \sum_{j=k+1}^{H-1} \sum_{\substack{(s_{k+1}, s_k, a_k) \in \mathcal{S}_{k+1} \times \mathcal{S}_k \times \mathcal{A} \\ (s_{j+1}, s_j, a_j) \in \mathcal{S}_{j+1} \times \mathcal{S}_j \times \mathcal{A}}} \frac{q_{[u]}^{\text{SEEDS-UT},P}(s_k, a_k)}{\max\left\{N_{[u]}(s_k, a_k)\right\}, 1\}} + \frac{q_{[u]}^{\text{SEEDS-UT},P}(s_j, a_j)}{\max\left\{N_{[u]}(s_j, a_j)\right\}, 1\}} \right).$$

Next, according to Cauchy-Schwarz inequality, we have the terms inside the big-$O$ notation can be upper-bounded by

$$\ln \frac{TSA}{\delta}$$

$$\cdot \left[ \sum_{k=0}^{H-1} \sum_{j=k+1}^{H-1} \sqrt{\sum_{u=1}^{\mathcal{U}} \sum_{\substack{(s_{k+1},s_k,a_k), \\ (s_{j+1},s_j,a_j)}} \frac{q_{[u]}^{\text{SEEDS-UT},P}(s_k,a_k) P(s_{k+1}|s_k,a_k) q_{[u]}^{\text{SEEDS-UT},P}(s_j,a_j|s_{k+1})}{\max\left\{N_{[u]}(s_k,a_k)\right\},1\right\}}}$$

$$\cdot \sqrt{\sum_{u=1}^{\mathcal{U}} \sum_{\substack{(s_{k+1},s_k,a_k), \\ (s_{j+1},s_j,a_j)}} \frac{q_{[u]}^{\text{SEEDS-UT},P}(s_k,a_k) P(s_{j+1}|s_j,a_j) q_{[u]}^{\text{SEEDS-UT},P}(s_j,a_j|s_{k+1})}{\max\left\{N_{[u]}(s_j,a_j)\right\},1\right\}}}$$

$$+ \sum_{u=1}^{\mathcal{U}} \sum_{k=0}^{H-1} \sum_{j=k+1}^{H-1} \sum_{\substack{(s_{k+1},s_k,a_k), \\ (s_{j+1},s_j,a_j)}} \left( \frac{q_{[u]}^{\text{SEEDS-UT},P}(s_k,a_k)}{\max\left\{N_{[u]}(s_k,a_k)\right\},1\right\}} + \frac{q_{[u]}^{\text{SEEDS-UT},P}(s_j,a_j)}{\max\left\{N_{[u]}(s_j,a_j)\right\},1\right\}} \right) \right].$$

Then, according to (34), we have that the terms under the $\sqrt{\cdot}$ operator can be upper-bounded by

$$\frac{1}{\tau} \sum_{u=1}^{\mathcal{U}} \sum_{\substack{(s_{k+1},s_k,a_k), \\ (s_{j+1},s_j,a_j)}} \sum_{t=(u-1)\tau+1}^{u\tau} \frac{q_{[u]}^{\text{SEEDS-UT},P}(s_k,a_k) P(s_{k+1}|s_k,a_k) q_{[u]}^{\text{SEEDS-UT},P}(s_j,a_j|s_{k+1})}{\max\left\{N_t(s_k,a_k)\right\},1\right\}}$$

$$\cdot \frac{1}{\tau} \sum_{u=1}^{\mathcal{U}} \sum_{\substack{(s_{k+1},s_k,a_k), \\ (s_{j+1},s_j,a_j)}} \sum_{t=(u-1)\tau+1}^{u\tau} \frac{q_{[u]}^{\text{SEEDS-UT},P}(s_k,a_k) P(s_{j+1}|s_j,a_j) q_{[u]}^{\text{SEEDS-UT},P}(s_j,a_j|s_{k+1})}{\max\left\{N_t(s_j,a_j)\right\},1\right\}},$$

and the second term in the bracket $[\cdot]$ can be upper-bounded by

$$\sum_{u=1}^{\mathcal{U}} \frac{1}{\tau} \sum_{k=0}^{H-1} \sum_{j=k+1}^{H-1} \sum_{\substack{(s_{k+1},s_k,a_k), \\ (s_{j+1},s_j,a_j)}} \sum_{t=(u-1)\tau+1}^{u\tau} \left( \frac{q_{[u]}^{\text{SEEDS-UT},P}(s_k,a_k)}{\max\left\{N_t(s_k,a_k)\right\},1\right\}} + \frac{q_{[u]}^{\text{SEEDS-UT},P}(s_j,a_j)}{\max\left\{N_t(s_j,a_j)\right\},1\right\}} \right).$$

Combining the above steps and according to Lemma 10 in Jin et al. (2020a), we have that the second term on the right-hand-side of (33) can be upper-bounded by $O\left(\frac{1}{\tau} H^2 S^2 A \ln \frac{TSA}{\delta}\right)$.

Therefore, with probability $1-\delta$, the first term on the right-hand-side of (26) can be upper-bounded by

$$O\left( HS\sqrt{AT \ln \frac{TSA}{\delta}} + H^2 S^2 \ln \frac{TSA}{\delta} \right). \tag{36}$$

*Step-2-ii (Bounding the second term):* The second term on the right-hand-side of (26) can be further decomposed into two terms as follows,

$$\sum_{u=1}^{\mathcal{U}} \mathbb{E}_{\mathcal{F}_{[u]}} \left[ \mathbb{E} \left[ \left\langle \hat{q}_{[u]}^{\text{SEEDS-UT},\mathcal{P}}, l_{[u]} - \hat{l}_{[u]}^{\text{SEEDS-UT}} \right\rangle \middle| \mathcal{F}_{[u]}, P \right] \right]$$

$$= \sum_{u=1}^{\mathcal{U}} \mathbb{E}_{\mathcal{F}_{[u]}} \left[ \mathbb{E} \left[ \left\langle \hat{q}_{[u]}^{\text{SEEDS-UT},\mathcal{P}}, l_{[u]} - \mathbb{E} \left[ \hat{l}_{[u]}^{\text{SEEDS-UT}} \right] \right\rangle \middle| \mathcal{F}_{[u]}, P \right] \right]$$

$$+ \sum_{u=1}^{\mathcal{U}} \mathbb{E}_{\mathcal{F}_{[u]}} \left[ \mathbb{E} \left[ \left\langle \hat{q}_{[u]}^{\text{SEEDS-UT},\mathcal{P}}, \mathbb{E} \left[ \hat{l}_{[u]}^{\text{SEEDS-UT}} \right] - \hat{l}_{[u]}^{\text{SEEDS-UT}} \right\rangle \middle| \mathcal{F}_{[u]}, P \right] \right]. \tag{37}$$

Let us consider the two terms on the right-hand-side. First, according to Lemma 2, we have

$$\sum_{u=1}^{\mathcal{U}} \mathbb{E}_{\mathcal{F}_{[u]}} \left[ \mathbb{E} \left[ \left\langle \hat{q}_{[u]}^{\text{SEEDS-UT},\mathcal{P}}, l_{[u]} - \mathbb{E} \left[ \hat{l}_{[u]}^{\text{SEEDS-UT}} \right] \right\rangle \middle| \mathcal{F}_{[u]}, P \right] \right]$$

$$= \sum_{u=1}^{\mathcal{U}} \mathbb{E}_{\mathcal{F}_{[u]}} \left[ \mathbb{E} \left[ \sum_{s\in\mathcal{S},a\in\mathcal{A}} \hat{q}_{[u]}^{\text{SEEDS-UT},\mathcal{P}}(s,a) l_{[u]}(s,a) \left( 1 - \frac{q_{[u]}^{\text{SEEDS-UT},P}(s,a)}{\mathcal{Q}_{[u]}^{\gamma}(s,a)} \right) \middle| \mathcal{F}_{[u]}, P \right] \right].$$

Since $l_{[u]}(s,a) \leq \tau$ and $\mathcal{Q}_{[u]}^{\gamma}(s,a) \geq \hat{q}_{[u]}^{\text{SEEDS-UT},\mathcal{P}}(s,a)$, we have

$$\sum_{u=1}^{\mathcal{U}} \mathbb{E}_{\mathcal{F}_{[u]}} \left[ \mathbb{E} \left[ \left\langle \hat{q}_{[u]}^{\text{SEEDS-UT},\mathcal{P}}, l_{[u]} - \mathbb{E} \left[ \hat{l}_{[u]}^{\text{SEEDS-UT}} \right] \right\rangle \middle| \mathcal{F}_{[u]}, P \right] \right]$$

$$\leq \tau \sum_{u=1}^{\mathcal{U}} \mathbb{E}_{\mathcal{F}_{[u]}} \left[ \mathbb{E} \left[ \sum_{s\in\mathcal{S},a\in\mathcal{A}} \left| \mathcal{Q}_{[u]}^{\gamma}(s,a) - q_{[u]}^{\text{SEEDS-UT},P}(s,a) \right| \middle| \mathcal{F}_{[u]}, P \right] \right]$$

$$\leq \tau \sum_{u=1}^{\mathcal{U}} \mathbb{E}_{\mathcal{F}_{[u]}} \left[ \mathbb{E} \left[ \sum_{s\in\mathcal{S},a\in\mathcal{A}} \left| \max_{\hat{P}\in\mathcal{P}_{[u]}} q_{[u]}^{\hat{P}}(s,a) + \gamma - q_{[u]}^{\text{SEEDS-UT},P}(s,a) \right| \middle| \mathcal{F}_{[u]}, P \right] \right],$$

where the term $\max_{\hat{P}\in\mathcal{P}_{[u]}} q_{[u]}^{\hat{P}}(s,a) - q_{[u]}^{\text{SEEDS-UT},P}(s,a)$ on the right-hand-side represents how well SEEDS-UT estimates the true occupancy measure using the transition-function set, and the term $\gamma$ on the right-hand-side verifies that this part of the gap is controlled by the parameter $\gamma$. Then, according to the bound for the first term on the right-hand-side of (26), we have

$$\sum_{u=1}^{\mathcal{U}} \mathbb{E}_{\mathcal{F}_{[u]}} \left[ \mathbb{E} \left[ \left\langle \hat{q}_{[u]}^{\text{SEEDS-UT},\mathcal{P}}, l_{[u]} - \mathbb{E} \left[ \hat{l}_{[u]}^{\text{SEEDS-UT}} \right] \right\rangle \middle| \mathcal{F}_{[u]}, P \right] \right]$$

$$\leq O \left( HS \sqrt{AT \ln \frac{TSA}{\delta}} \right) + \gamma TSA.$$

Second, according to Azuma's inequality, we have with probability $1 - \delta$,

$$\sum_{u=1}^{\mathcal{U}} \mathbb{E}_{\mathcal{F}_{[u]}} \left[ \mathbb{E} \left[ \left\langle \hat{q}_{[u]}^{\text{SEEDS-UT},\mathcal{P}}, \mathbb{E} \left[ \hat{l}_{[u]}^{\text{SEEDS-UT}} \right] - \hat{l}_{[u]}^{\text{SEEDS-UT}} \right\rangle \middle| \mathcal{F}_{[u]}, P \right] \right]$$

$$\leq O \left( \tau H \sqrt{\frac{T}{\tau} \ln \frac{1}{\delta}} \right) \leq O \left( H \sqrt{T\tau \ln \frac{1}{\delta}} \right). \tag{38}$$

Therefore, with probability $1 - \delta$, the second term on the right-hand-side of (26) can be upper-bounded by

$$O \left( HS \sqrt{AT \ln \frac{TSA}{\delta}} + \gamma TSA + H \sqrt{T\tau \ln \frac{1}{\delta}} \right). \tag{39}$$

*Step-2-iii (Bounding the third term):* Follow our proof for the case when the transition function is known, it is not hard to show that

$$\sum_{u=1}^{\mathcal{U}} \left\langle \hat{q}_{[u]}^{\text{SEEDS-UT},\mathcal{P}} - q^{\pi^*}, \hat{l}_{[u]}^{\text{SEEDS-UT}} \right\rangle$$

$$\leq \eta \sum_{u=1}^{\mathcal{U}} \sum_{s\in\mathcal{S},a\in\mathcal{A}} \hat{q}_{[u]}^{\text{SEEDS-UT},\mathcal{P}}(s,a) \left( \hat{l}_{[u]}^{\text{SEEDS-UT}}(s,a) \right)^2 + \frac{H \ln(SA)}{\eta}.$$

Let us focus on the first term on the right-hand-side. Note that different from that in Jin et al. (2020a), the loss $\hat{l}_{[u]}^{\text{SEEDS-UT}}(s,a)$ above is calculated based on the samples from a whole super-episode. Thus,

each state-action pair could be visited multiple times. To this end, we provide a super-episodic version of loss concentration as follows,

$$\sum_{u=1}^{\mathcal{U}} \sum_{s \in \mathcal{S}, a \in \mathcal{A}} \hat{q}_{[u]}^{\text{SEEDS-UT},\mathcal{P}}(s,a) \left( \hat{l}_{[u]}^{\text{SEEDS-UT}}(s,a) \right)^2 \leq \frac{\tau H}{2\gamma} \ln \frac{H}{\delta}$$

$$+ \sum_{u=1}^{\mathcal{U}} \sum_{s \in \mathcal{S}, a \in \mathcal{A}} \frac{\tau q_{[u]}^{\text{SEEDS-UT}}}{\max\limits_{\hat{P} \in \mathcal{P}_{[u]}} q_{[u]}^{\hat{P}}(s,a)} l_{[u]}(s,a).$$

In the following, we show how to get this. First, since

$$\hat{l}_{[u]}^{\text{SEEDS-UT}}(s,a) = \sum_{j=1}^{J_{[u]}} \frac{l_{t_j(s,a)}(s,a)}{\mathcal{Q}_{[u]}^{\gamma}(s,a)} \mathbf{1}_{\{(s,a) \text{ was visited in episodes } t_1(s,a),\ldots,t_{J_{[u]}}(s,a) \text{ of super-episode } u\}}$$

$$\leq \frac{\tau}{\mathcal{Q}_{[u]}^{\gamma}(s,a)},$$

we have

$$\hat{q}_{[u]}^{\text{SEEDS-UT},\mathcal{P}}(s,a) \left( \hat{l}_{[u]}^{\text{SEEDS-UT}}(s,a) \right)^2 \leq \frac{\tau \hat{q}_{[u]}^{\text{SEEDS-UT},\mathcal{P}}(s,a)}{\mathcal{Q}_{[u]}^{\gamma}(s,a)} \hat{l}_{[u]}^{\text{SEEDS-UT}}(s,a)$$

$$\leq \tau \hat{l}_{[u]}^{\text{SEEDS-UT}}(s,a)$$

$$= \tau \sum_{j=1}^{J_{[u]}} \frac{l_{t_j(s,a)}(s,a)}{\mathcal{Q}_{[u]}^{\gamma}(s,a)} \mathbf{1}_{\{(s,a) \text{ was visited in episodes } t_1(s,a),\ldots,t_{J_{[u]}}(s,a) \text{ of super-episode } u\}}$$

$$= \tau \sum_{t=(u-1)\tau+1}^{u\tau} \frac{l_t(s,a)}{\mathcal{Q}_{[u]}^{\gamma}(s,a)} \mathbf{1}_{\{(s,a) \text{ was visited in episode } t \text{ of super-episode } u\}}.$$

Let us define

$$\tilde{l}_t(s,a) \triangleq \frac{l_t(s,a)\mathbf{1}_{\{(s,a) \text{ was visited in episode } t \text{ of super-episode } u\}}}{\mathcal{Q}_{[u]}^{\gamma}(s,a)}.$$

Then, we have

$$\sum_{t=1}^{T} \sum_{s \in \mathcal{S}, a \in \mathcal{A}} 2\gamma \left( \tilde{l}_t(s,a) - \frac{q_{[u]}^{\text{SEEDS-UT}}}{\max\limits_{\hat{P} \in \mathcal{P}_{[u]}} q_{[u]}^{\hat{P}}(s,a)} l_t(s,a) \right) \leq H \ln \frac{H}{\delta}.$$

By combining all episodes in the same super-episode $u$ together, we have

$$\sum_{u=1}^{\mathcal{U}} \sum_{s \in \mathcal{S}, a \in \mathcal{A}} 2\gamma \left( \sum_{t=(u-1)\tau+1}^{u\tau} \tilde{l}_t(s,a) - \frac{q_{[u]}^{\text{SEEDS-UT}}}{\max\limits_{\hat{P} \in \mathcal{P}_{[u]}} q_{[u]}^{\hat{P}}(s,a)} l_{[u]}(s,a) \right) \leq H \ln \frac{H}{\delta}.$$

By rearranging the terms, we have

$$\sum_{u=1}^{\mathcal{U}} \sum_{s \in \mathcal{S}, a \in \mathcal{A}} \tilde{l}_{[u]}(s,a) \leq \frac{H}{2\gamma} \ln \frac{H}{\delta} + \sum_{u=1}^{\mathcal{U}} \sum_{s \in \mathcal{S}, a \in \mathcal{A}} \frac{q_{[u]}^{\text{SEEDS-UT}}}{\max\limits_{\hat{P} \in \mathcal{P}_{[u]}} q_{[u]}^{\hat{P}}(s,a)} l_{[u]}(s,a)$$

$$\leq \frac{H}{2\gamma} \ln \frac{H}{\delta} + \sum_{u=1}^{\mathcal{U}} \sum_{s \in \mathcal{S}, a \in \mathcal{A}} l_{[u]}(s,a) \leq \frac{H}{2\gamma} \ln \frac{H}{\delta} + \frac{T}{\tau} SA\tau = \frac{H}{2\gamma} \ln \frac{H}{\delta} + TSA.$$

Thus, we have

$$\sum_{u=1}^{\mathcal{U}} \sum_{s \in \mathcal{S}, a \in \mathcal{A}} \hat{q}_{[u]}^{\text{SEEDS-UT},\mathcal{P}}(s,a) \left( \hat{l}_{[u]}^{\text{SEEDS-UT}}(s,a) \right)^2 \leq \tau \cdot \frac{H}{2\gamma} \ln \frac{H}{\delta} + \tau TSA.$$

Therefore, with probability $1 - \delta$, the third term on the right-hand-side of (26) can be upper-bounded by

$$O\left(\frac{\eta\tau H}{\gamma}\ln\frac{H}{\delta} + \eta\tau TSA + \frac{H\ln(SA)}{\eta}\right).\tag{40}$$

*Step-2-iv (Bounding the fourth term):* First, it is not hard to get that with probability $1 - \delta$,

$$\sum_{u=1}^{\mathcal{U}}\hat{l}_{[u]}^{\text{SEEDS-UT}}(s,a) \leq \frac{1}{2\gamma}\ln\frac{H}{\delta} + \sum_{u=1}^{\mathcal{U}}\frac{q_{[u]}^{\text{SEEDS-UT},P}(s,a)}{\max\limits_{\hat{P}\in\mathcal{P}_{[u]}}q_{[u]}^{\hat{P}}(s,a)}l_{[u]}(s,a).\tag{41}$$

Thus, we have

$$\sum_{u=1}^{\mathcal{U}}\left\langle q^{\pi^*}, \hat{l}_{[u]} - l_{[u]}\right\rangle = \sum_{u=1}^{\mathcal{U}}\sum_{s\in\mathcal{S},a\in\mathcal{A}}q^{\pi^*}(s,a)\hat{l}_{[u]}(s,a) - \sum_{u=1}^{\mathcal{U}}\sum_{s\in\mathcal{S},a\in\mathcal{A}}q^{\pi^*}(s,a)l_{[u]}(s,a)$$

$$\leq \sum_{s\in\mathcal{S},a\in\mathcal{A}}q^{\pi^*}(s,a)\frac{1}{2\gamma}\ln\frac{H}{\delta} + \sum_{s\in\mathcal{S},a\in\mathcal{A}}q^{\pi^*}(s,a)\cdot\sum_{u=1}^{\mathcal{U}}\frac{q_{[u]}^{\text{SEEDS-UT},P}(s,a)}{\max\limits_{\hat{P}\in\mathcal{P}_{[u]}}q_{[u]}^{\hat{P}}(s,a)}l_{[u]}(s,a)$$

$$- \sum_{u=1}^{\mathcal{U}}\sum_{s\in\mathcal{S},a\in\mathcal{A}}q^{\pi^*}(s,a)l_{[u]}(s,a).$$

$$\leq \frac{H}{2\gamma}\ln\frac{H}{\delta} + \sum_{u=1}^{\mathcal{U}}\sum_{s\in\mathcal{S},a\in\mathcal{A}}q^{\pi^*}(s,a)l_{[u]}(s,a)\left(\frac{q_{[u]}^{\text{SEEDS-UT},P}(s,a)}{\max\limits_{\hat{P}\in\mathcal{P}_{[u]}}q_{[u]}^{\hat{P}}(s,a)} - 1\right)$$

$$\leq \frac{H}{2\gamma}\ln\frac{H}{\delta}.\tag{42}$$

**Step-3 (Final step):** Finally, by combining (36), (39), (40), (42) and the switching-cost upper-bound $\beta\cdot\left\lceil\frac{T}{\tau}\right\rceil$, and tuning the parameters $\eta$, $\tau$ and $\gamma$ as in Algorithm 2, we have that the regret of SEEDS-UT is upper-bounded by $O\left(\beta^{1/3}H^{2/3}(SA)^{1/3}T^{2/3}\left(\ln\frac{TSA}{\delta}\right)^{1/2}\right)$ with probability $1 - \delta$.

$\square$

