# OpenReview forum: "Near-Optimal Adversarial Reinforcement Learning with Switching Costs"
_ICLR.cc/2023/Conference — ICLR 2023 notable top 25%_

### Official Review · Reviewer_fZFZ · 2022-10-23

**Confidence:** 3
**Correctness:** 3
**Technical Novelty And Significance:** 3
**Empirical Novelty And Significance:** Not applicable
**Recommendation:** 8

**Clarity, Quality, Novelty And Reproducibility:**

As I discussed for the last question, the main body is written clearly but the writing in the proofs needs to be improved. As for the novelty, I feel the most original part is the design of SEEDS algorithm. In contrast, the lower bound result and the extension to the unknown transition probability (using the empirical estimators) are small modifications based on previous works.

**Strength And Weaknesses:**

Strength:

This paper makes solid contribution to adversarial RL with switching costs. While the regret lower bound result is not very surprising because similar lower bounds have been shown for the special case of adversarial bandits, it is good to see that the regret upper bound can match the lower bound exactly in terms of $T, H, S, A, \beta$ (episode number, episode length, state space size, action space size, and switching cost weight) when the transition probabilities are known. The authors also make their algorithm more practical by replacing the exact probability with an empirical estimator.

The paper is well-written and easy to follow. The authors add adequate discussions about the inspiration from previous works, and state the major technical difficulties clearly.

Weakness:

1.The proofs are not written clear enough for me to verify the results. For example, in the proof of Theorem 1 in Appendix A, what is the transition probability in this example? And what do you mean by the order of states?

2.It is not clear whether the proposed algorithms can be implemented efficiently, because the authors do not elaborate on how to compute equation (10). I conjecture the complexity will be high because the set $C(P)$ can be complicated given that it needs to consider all possible policies. This issue might become worse when the transition probability is unknown, because the estimation of transition probability is keep changing.

3.I feel the problem setting of adversarial RL with switching costs is closely related to online control, but that literature is not included in “Related Works”. Specifically, I feel the stronger regret performance metrics in that literature, like the dynamic regret or the adaptive regret, can work better than static regret in the non-static environments like adversarial RL. I recommend the authors to add a discussion in revision.


**Summary Of The Paper:**

This paper studies adversarial reinforcement learning (RL) with switching costs in an episodic setting, where a constant switching cost is incurred whenever the agent deploys a different policy. The authors first show a $\Omega(T^{2/3})$ regret lower bound, which generalizes the regret lower bound for adversarial Multi-Armed Bandits (MAB) with switching costs. Then, they propose two algorithms, SEEDS and SEEDS-UT, which can achieve provable regret guarantees. The first algorithm, SEEDS, requires the exact knowledge of transition probabilities and can achieve a regret that matches the regret lower bound. The second algorithm, SEEDS-UT, does not require the knowledge of transition probabilities. Its regret almost matches the lower bound with a gap of $H^{1/3}$, where $H$ is the length of each episode.

**Summary Of The Review:**

In summary, this paper makes solid contribution to adversarial RL with switching costs by providing the first matching regret upper and lower bounds. Compared to the significance of the contributions, the weakness I mentioned are not major concerns. Therefore, I recommend for accept.

Since the proofs are not carefully checked due to presentation issues and time limit, I prefer to choose a low confidence level.

---

> ### Author Response · Authors · 2022-11-15
> **Response to Reviewer fZFZ**
>
> We thank the reviewer for providing the helpful and positive review! We have addressed the reviewer’s helpful comments and modified the paper accordingly. Please note that in the revised paper, we highlighted our changes by blue-colored texts in both the main body and the appendices of the paper.
>
> **Q1 [Weakness 1]:** The proofs are not written clearly enough for me to verify the results. For example, in the proof of Theorem 1 in Appendix A, what is the transition probability in this example? And what do you mean by the order of states?
>
> **A1:** Thanks for the question. In the proof of Theorem 1 in Appendix A, we constructed deterministic state transitions. Specifically, as we stated at the end of the second paragraph in Appendix A, the state transition is that the $i$-th state at a layer will always transit to the $i$-th state at the next layer, i.e., $P(s_i|s_i,a)=1$ for all $a$, and $P(s_j|s_i,a)=0$ for all $j \neq i$ and $a$. Moreover, the index $i$ is the order of the state.
>
> In the revised paper, we have provided more details for both the transition probabilities and the definition of the order of states in Appendix A.
>
> **Q2 [Weakness 2]:** It is not clear whether the proposed algorithms can be implemented efficiently, because the authors do not elaborate on how to compute equation (10). I conjecture the complexity will be high because the set $\mathbb{C}(P)$ can be complicated given that it needs to consider all possible policies. This issue might become worse when the transition probability is unknown, because the estimation of transition probability keeps changing.
>
> **A2:** Thanks for the comments. We clarify that the proposed algorithms can be implemented efficiently. Due to page limits, the details for this part are included in appendices. Specifically, as we mentioned below Eq. (4), $\mathbb{C}(P)$ is formulated by Eqs. (2)-(4). Thus, (10) is a simple constrained convex optimization problem. In Appendix D.1, we provided the details about how to solve the unconstrained version using the standard convex optimization method, and then the constrained version can be solved similarly. More specifically, in Eq. (20), we provided the explicit form of the solution to  the unconstrained problem, which is equal to the last occupancy measure times an exponential term. The final solution of Eq. (10) will then be equal to the last occupancy measure times a slightly different exponential term. To see this, we can substitute constraints (2)-(4) to the objective function using Lagrangian multipliers, and then solve it using the standard convex optimization method. Similarly, when the transition probability is unknown, $\mathbb{C}(P)$ is formulated by Eqs. (2)-(4) together with Eq. (14). Then, the problem can be solved similarly.
>
> In the revised paper, we have added ``Recall that $\mathbb{C}(P)$ is formulated by (2)-(4).’’ after Eq. (10). Moreover, we have added the explicit form of the final solution to Eq. (10) after Eq. (20).
>
> **Q3: [Weakness 3]:** I feel the problem setting of adversarial RL with switching costs is closely related to online control, but that literature is not included in “Related Works”. Specifically, I feel the stronger regret performance metrics in that literature, like the dynamic regret or the adaptive regret, can work better than static regret in the non-static environments like adversarial RL. I recommend the authors to add a discussion in revision.
>
> **A3:** Thank you for the suggestions! In the revised paper, we have added related work on online control in the related-work section (i.e., Sec. 2). Further, we have added the dynamic regret as a topic for future work in the conclusion (i.e., Sec. 7).
>
>
>
> Finally, we thank the reviewer again for the helpful comments and suggestions for our work. We are more than happy to address any further questions that you may have during the discussion period.

---

### Official Review · Reviewer_Vm2r · 2022-10-23

**Confidence:** 2
**Correctness:** 3
**Technical Novelty And Significance:** 3
**Empirical Novelty And Significance:** Not applicable
**Recommendation:** 8

**Clarity, Quality, Novelty And Reproducibility:**

This paper is well-written and easy to follow. However, the lower bound in Theorem 1 is just a simple extension from bandit to MDP. The novelty of the lower bound seems limited.

**Strength And Weaknesses:**

Strength:

The author provides the first analyses on switching costs in adversarial RL, and proposed novel algorithms SEEDs with $O(T^{2/3})$ regret guarantee for both known and unknown transitions. In addition, the theoretical lower bound shows that the proposed SEEDs algorithms are near-optimal.

Weakness:

Some of the paper’s claims need to be clarified.

1. The theoretical lower bound in Theorem 1 highly depends on the switching coefficient $\beta$, and when $\beta$ is small (e.g., $0$ or $O(1/T)$), the lower bound is smaller than $O(T^{2/3})$. Thus, it seems necessary to indicate the coefficient $\beta$ in the abstract and introduction.

2. The theoretical lower bound in Theorem 1 requires the number of episodes $T$ to be large enough. Otherwise, the lower bound is larger than $\Omega(T)$.

3. For the related work, there exist different assumptions for the MDP. More specifically, the adversarial MDP assumes the state in each stage $h$ is non-overlap, and static MDP does not make this assumption. Therefore, it is better to indicate this difference for a fair comparison between related works.

**Summary Of The Paper:**

This work focused on adversarial MDPs with switching costs. The author proposed the SEEDs algorithms and provided a $O(T^{2/3})$ regret guarantee for both known and unknown transitions. In addition, the author also provided a theoretical guarantee for the lower regret bound, which suggests the proposed SEEDs algorithms are near-optimal on adversarial MDPs with switching costs.

**Summary Of The Review:**

The author provides the first analyses on switching costs in adversarial RL, and proposed novel algorithms SEEDs with $O(T^{2/3})$ regret guarantee for both known and unknown transitions. In addition, the theoretical lower bound shows that the proposed SEEDs algorithms are near-optimal. Although some of the paper’s claims need to be clarified, they can be easily fixed with minor changes.

---

> ### Author Response · Authors · 2022-11-15
> **Response to Reviewer Vm2r**
>
> We thank the reviewer for providing the helpful and positive review! We have addressed the reviewer’s helpful comments and modified the paper accordingly. Please note that in the revised paper, we highlighted our changes by blue-colored texts in both the main body and the appendices of the paper.
>
> **Q1 [Weakness 1]:** The theoretical lower bound in Theorem 1 highly depends on the switching coefficient $\beta$, and when $\beta$ is small (e.g., $0$ or $O(1/T)$), the lower bound is smaller than $O(T^{2/3})$. Thus, it seems necessary to indicate the coefficient $\beta$ in the abstract and introduction.
>
> **A1:** Thank you for the suggestion! Indeed, the switching-cost coefficient $\beta$ is strictly positive and is independent of $T$. This definition follows the standard definition of the switching-cost coefficient in both RL and bandits, e.g., Bai et al. (2019), Wang et al. (2021), Qiao et al. (2022), and Shi et al. (2022).
>
> As suggested by the reviewer, in the revised paper, we have added “(with a coefficient $\beta$ that is strictly positive and is independent of $T$)” in the abstract and in the first sentence of the third paragraph in the introduction (i.e., Sec. 1).
>
> References:
>
> Yu Bai, Tengyang Xie, Nan Jiang, and Yu-Xiang Wang. Provably efficient q-learning with low switching cost. Advances in Neural Information Processing Systems, 32, 2019.
>
> Tianhao Wang, Dongruo Zhou, and Quanquan Gu. Provably efficient reinforcement learning with linear function approximation under adaptivity constraints. Advances in Neural Information Processing Systems, 34, 2021.
>
> Dan Qiao, Ming Yin, Ming Min, and Yu-Xiang Wang. Sample-efficient reinforcement learning with $\log \log(t)$ switching cost. arXiv preprint arXiv:2202.06385, 2022.
>
> Ming Shi, Xiaojun Lin, and Lei Jiao. Power-of-2-arms for bandit learning with switching costs. In 23rd International Symposium on Theory, Algorithmic Foundations, and Protocol Design for Mobile Networks and Mobile Computing (ACM MobiHoc). pp. 131-140, 2022.
>
> **Q2 [Weakness 2]:** The theoretical lower bound in Theorem 1 requires the number of episodes $T$ to be large enough. Otherwise, the lower bound is larger than $\Omega(T)$.
>
> **A2:** Many thanks. Yes, $T$ should be large enough. In the revised paper, we have added “and $T\geq\max{\\{6H^{2}SA,\beta\\}}$” in Theorem 1.
>
> **Q3 [Weakness 3]:** For the related work, there exist different assumptions for the MDP. More specifically, the adversarial MDP assumes the state in each stage $h$ is non-overlap, and static MDP does not make this assumption. Therefore, it is better to indicate this difference for a fair comparison between related works.
>
> **A3:** Thanks for pointing this out. Yes, we agree. In the revised paper, in the third paragraph of the related-work section (i.e., Sec. 2), we have added the following sentence: “These studies assume that the state spaces of layers in an episode are non-overlapping.”
>
>
>
> Finally, we thank the reviewer again for the helpful comments and suggestions for our work. We are more than happy to address any further questions that you may have during the discussion period.

---

### Official Review · Reviewer_TSyk · 2022-10-24

**Confidence:** 4
**Clarity, Quality, Novelty And Reproducibility:** Please find my comments in the above …
**Correctness:** 4
**Technical Novelty And Significance:** 2
**Empirical Novelty And Significance:** Not applicable
**Recommendation:** 6

**Strength And Weaknesses:**

Strengths:
 * The topic of the paper should be of good interest to the community. The paper focuses on Adversarial MDPs (AMDPs), an MDP with fixed (unknown) transition and adversarial reward/loss functions, which can be viewed as a tractable adversarial analog of standard stochastic MDPs. Studying the switching cost problem has been done in both adversarial bandits, and stochastic MDPs. Thus a first result on switching cost in adversarial MDPs is definitely of interest.

* The obtained result of $\widetilde{O}(T^{2/3})$ regret and switching cost is solid, matches the result in adversarial bandits and slightly worse than in stochastic MDPs, which are expected. The result is established in both the known transition setting, and the unknown transition setting (which in my opinion is the main interesting one).

* The paper is extremely clearly presented and quite easy to follow, with many intuitions discussed in detail and the related work adequately discussed. The paper could be a good reference point in the future for AMDPs and switching costs in adversarial settings.


Weaknesses:

* The main concern is that, at the heart, the technical contribution of this paper may be a bit incremental—The entire algorithm seems to be a direct combination of the standard way to do adversarial bandits with switching costs with $T^{2/3}$ regret using the “batched” EXP3 algorithm, and the algorithm of Jin et al. (2020a) to learn AMDPs with unknown transitions. I feel the combination is direct, because this “batched EXP3 algorithm” for bandits can be done with fixed, equally-sized batches, and thus when applied in AMDPs, across different states, the policy switches can be straightforwardly synced, as done in Algorithm 1 & 2.
In particular, this is comparatively easier than logarithmic switching cost bounds in the stochastic MDP setting, where the low switching mechanism is an exponential grid depending over the visitation count, thus has to be done asynchronously for each state.

* Related to the above concern, I feel like the presentation of many technical intuitions are slightly over-repetitive and perhaps a bit overclaiming. As an example, the “novel idea for estimating the loss” part on Page 6, the provided baseline is indeed not quite sensible and non-standard, as the losses change in each episode. The provided solution instead, in Eq(8), is just averaging of importance-weighted loss estimator in standard EXP3, and seems a natural/standard practice to me instead of a really novel idea. On this end, I think it’s probably good to shorten the discussions of these standard practices, and expand on the more nontrivial bits (for example, an explanation of how the $T^{2/3}$ regret is achievable in bandits for readers not familiar with that, and/or how that integrates with the AMDP algorithm of Jin et al. (2020a) to give the final result).


**Summary Of The Paper:**

This paper studies the problem of online reinforcement learning with adversarial losses and switching costs, where there is a cost for switching the policy during learning. The main result is an algorithm that achieves $\tilde{O}(T^{2/3})$ regret and switching cost, for both the setting of known and unknown transitions. The paper also provides a matching $\Omega(T^{2/3})$ lower bound.

**Summary Of The Review:**

Overall, the paper obtains solid and interesting results for learning AMDPs with low switching cost, and is very clearly presented. However, I feel like technically it’s a bit incremental and unclear how the proof techniques could inspire future work.

---

> ### Author Response · Authors · 2022-11-15
> **Response to Reviewer TSyk (part 2)**
>
> **Q2 [Weakness 2]:** Related to the above concern, I feel like the presentation of many technical intuitions are slightly over-repetitive and perhaps a bit overclaiming. As an example, the "novel idea for estimating the loss" part on Page 6, the provided baseline is indeed not quite sensible and non-standard, as the losses change in each episode. The provided solution instead, in Eq. (8), is just averaging of importance-weighted loss estimator in standard EXP3, and seems a natural/standard practice to me instead of a really novel idea. On this end, I think it’s probably good to shorten the discussions of these standard practices, and expand on the more nontrivial bits (for example, an explanation of how the $T^{2/3}$ regret is achievable in bandits for readers not familiar with that, and/or how that integrates with the AMDP algorithm of Jin et al. (2020a) to give the final result).
>
> **A2:** Many thanks for the comments.
>
> * First, we felt that our baseline can be a natural estimator from an easy-to-analyze perspective.  We agree with the reviewer that we can shorten it to leave space for some other material.
>
> * Second, we clarify that Eq. (8) is not just the average of importance-weighted loss estimator in standard Exp3, because the denominator on the right-hand-side of Eq. (8) is neither the probability of visitation in the super-episode (which should be $1-(1-q_{[u]})^{\tau}$ instead), nor the probability of visitation in each episode of the super-episode (which should be $q_t$ for the first episode and $1$ for all subsequent episodes). Instead, the denominator on the right-hand-side of Eq. (8) is constructed to be the occupancy measure updated at the end of the last super-episode. This new design provides the unbiased estimated loss as we have shown in Lemma 3, while simply averaging the standard Exp3 importance-weighted loss estimator will not. Moreover, as we discuss in A1, this new design results in new challenges in the proof of Lemma 3 (which is critical for the final regret proof of Theorem 4). To this end, we established a critical property, which shows that under this new design, summing over all possible sets of the random episodes where the state-action pair was visited is equivalent to summing over all deterministic episodes from the beginning to the end of a super-episode.
>
> * Third, as the reviewer suggested, we have revised our presentation focus in the revision. Specifically, we have added a new paragraph to show how the $T^{2/3}$ regret is achievable in bandits, and we have rephrased some sentences in Sec. 6 and the proof of Theorem 6 to better emphasize the new analysis developments when extending to the general case.
>
>
>
> Finally, we thank the reviewer again for the helpful comments and suggestions for our work. We are more than happy to address any further questions that you may have during the discussion period.

---

> ### Author Response · Authors · 2022-11-15
> **Response to Reviewer TSyk (part 1)**
>
> We thank the reviewer for providing the helpful and positive review! We have addressed the reviewer’s helpful comments and modified the paper accordingly. Please note that in the revised paper, we highlighted our changes by blue-colored texts in both the main body and the appendices of the paper.
>
> **Q1 [Weakness 1]:** The main concern is that, at the heart, the technical contribution of this paper may be a bit incremental-The entire algorithm seems to be a direct combination of the standard way to do adversarial bandits with switching costs with $T^{2/3}$ regret using the "batched" EXP3 algorithm, and the algorithm of Jin et al. (2020a) to learn AMDPs with unknown transitions. I feel the combination is direct, because this "batched EXP3 algorithm" for bandits can be done with fixed, equally-sized batches, and thus when applied in AMDPs, across different states, the policy switches can be straightforwardly synced, as done in Algorithm 1 & 2. In particular, this is comparatively easier than logarithmic switching cost bounds in the stochastic MDP setting, where the low switching mechanism is an exponential grid depending over the visitation count, thus has to be done asynchronously for each state.
>
> **A1:** We respectfully disagree that our algorithm is a direct combination of those standard methods. First of all, we understand that in **stochastic** MDP, the nature of the problem calls for an asynchronous design for each state in order to achieve logarithmic switching cost bounds. However, the challenge in the **adversarial** MDP here is different. Our lower bounds in Theorem 1 and Theorem 2 have suggested that, differently from stochastic MDP, with a logarithmic switching cost, it is not possible to achieve a sub-linear loss regret (e.g., $O(\sqrt{T})$ in stochastic MDP). Thus, we did not adopt an asynchronous design, but instead chose to have equal-sized super-episodes. On the other hand, we still need to handle new challenges arising in adversarial MDP (which do not exist in stochastic MDP). Due to the arbitrarily changing loss function, our algorithms have several new design components that warrant merits. Further, these new design ingredients require new technical analysis as well. We now highlight the key differences below.
>
> - **(a)** Compared with batched Exp3, our design of the estimated loss in Eq. (8) to address the random visitations of the state-action pairs in a super-episode is different and new. (Recall that in batched Exp3 for bandits, there is no such difficulty.) This new design poses new challenges to establish that the new loss estimator is unbiased in the proof of Lemma 3, which is critical for the final regret proof of Theorem 4. To this end, we established a critical property, which shows that under this new design, summing over all possible sets of the random episodes where the state-action pair was visited is equivalent to summing over all deterministic episodes from the beginning to the end of a super-episode.
>
> - **(b)** Differently from the algorithm of Jin et al. (2020a), our algorithm 2 estimates the transition function based on all samples collected from the whole super-episode. This new design poses new challenges to bound the regret over the super-episode, where each state-action pair could be visited multiple times, and such visitations are random. In contrast, in Jin et al. (2020a), each state-action pair is assumed to be visited only once in an episode, and clearly their proof is not applicable here. To resolve this difficulty, in Eq. (35) of Appendix G, we carefully constructed a convergent series based on the collected samples to achieve an analyzable intermediate step for our proof of Theorem 6. Specifically, this special series is used to bound the estimation error of the updated occupancy measure and the estimated loss, which are functions of the estimated transition function based on the samples collected from the whole super-episode, in step-2-i and step-2-iv in Appendix G, respectively.
>
> Reference:
> Chi Jin, Tiancheng Jin, Haipeng Luo, Suvrit Sra, and Tiancheng Yu. Learning adversarial markov decision processes with bandit feedback and unknown transition. In International Conference on Machine Learning, pp. 4860–4869. PMLR, 2020a.

---

### Official Review · Reviewer_y6Wb · 2022-10-25

**Confidence:** 4
**Correctness:** 2
**Technical Novelty And Significance:** 2
**Empirical Novelty And Significance:** Not applicable
**Recommendation:** 6

**Clarity, Quality, Novelty And Reproducibility:**

Quality: The proof in this new setting is correct.

Clarity: This work is well-organized.

Originality: This work is the first to deal with the switching cost in adversarial RL. However, the proof process is similar to the previous works. Hence I think this work represents incremental advances in a new setting.


**Strength And Weaknesses:**

Strengths:

This work is the first to deal with the switching cost in adversarial RL. The algorithm in this paper (almost) matches the lower bound if the lower bound is correct.

Weakness：

1. The definition of the switching cost is unreasonable. In this paper, as long as the policy distribution changes slightly, there will be a switching cost. Imagine that if $\pi_{t+1}$ and $\pi_{t}$ are only slightly different (for example, a small KL divergence), we can fine-tune the policy. Then we can use offline evaluation methods such as importance sampling to evaluate the new policy, and there is no need to pay a high cost.

2. The analysis of the lower bound in this paper is confusing. As stated in Weakness 1, the indicator function consider whether the policy distribution $\pi_{t+1}$ and $\pi_{t}$ are different, not $a_{t+ 1}$ and $a_{t}$ are the same. However, for adversarial bandits with switching costs, the indicator function considers whether $a_{t+1}$ and $a_{t}$ are different. These are two completely different measures, so the lower bound proved by Shi et al. (2022) cannot be used directly.

3. I did not see too many difficulties in the regret analysis process of Theorem 4. It seems that the proof process of Theorem 4 is a standard online mirror descent (OMD) proof process, except that super-episodes replace episodes. It is worth describing the non-trivial tricks of the proof process.

4. The core components of extending known transition to unknown transition are exactly the same as Jin et al. [2].

[1] Shi, M., Lin, X., & Jiao, L. (2022, October). Power-of-2-arms for bandit learning with switching costs. In Proceedings of the Twenty-Third International Symposium on Theory, Algorithmic Foundations, and Protocol Design for Mobile Networks and Mobile Computing (pp. 131-140).

[2] Chi Jin, Tiancheng Jin, Haipeng Luo, Suvrit Sra, and Tiancheng Yu. Learning adversarial markov decision processes with bandit feedback and unknown transition. In International Conference on Machine Learning, pp. 4860–4869. PMLR, 2020.

Question:

Q1: Can you explain in detail why you can use the lower bound proved by Shi et al. (2022) directly?

Q2: Can you propose a more reasonable way to measure the difference between the two policy distributions and prove an upper bound?

----

It seems that the authors indeed work with deterministic policies. Then the setting and lower bound make sense to me. Though I still feel that the techniques are a bit incremental, it is not a fundamental issue. I would like to increase my score to 6.

**Summary Of The Paper:**

This work studies adversarial reinforcement learning with switching costs in the tabular setting. This work designs the super-episode containing $\tau$ episodes and proposes an occupancy measure-based algorithm SEEDS. To control the switching cost, SEEDS uses the same policy in a super-episode and only updates the occupancy measure at the last of each super-episode. As the first step, this work proves the lower bound $\tilde{\Omega}((HSA)^{1/3}T^{2/3})$ in this setting. By carefully choosing $\tau$ to balance the loss regret and the switching cost, this work proves SEEDS achieve $\tilde{O}((HSA)^{1/3}T^{2/3})$ regret guarantee which matches the lower bound with the known transition. With the unknown transition, SEEDS-UT almost matches the lower bound within a factor of $\tilde{O}(H^{1/3})$.

**Summary Of The Review:**

This work proposes an occupancy measure-based algorithm SEEDS using super-episodes and (almost) matches the lower bound. However, the definition of the switching cost is not very reasonable. Furthermore, the proof of the lower bound is a bit confusing. The lower bound proved by the previous work can not be directly used in this work. Finally, the technique when proving the upper bound is almost the same as the standard proof process of online mirror descent and the proof process of the previous work [2].

---

> ### Author Response · Authors · 2022-11-15
> **Response to Reviewer y6Wb (part 3)**
>
> **Q6 [Weakness 4]:** The core components of extending known transition to unknown transition are exactly the same as Jin et al. (2020a).
>
> **A6:** We respectfully disagree with the above comment. Generally speaking, since our algorithm introduces super-episodes that do not exist in Jin et al. (2020a), all the technical complications that we need to handle due to such super-episodes do not show up in Jin et al. (2020a). Specifically, our extension involves several key new components as follows.
>
> - **(a)** Due to the delayed switching of our SEEDS-UT algorithm (i.e., Algorithm 2 for  the unknown transition case), our algorithm collects samples from a whole super-episode to estimate the transition function, and hence each state-action pair could be visited multiple times, and such visitations are random. In contrast, in Jin et al. (2020a), each state-action pair is assumed to be visited at most once in an episode, which is required in many steps (e.g., Lemma 4 and Lemma 10) of their proof. Thus, the proof from Jin et al. (2020a) is not applicable in our case. To resolve this difficulty, in Eq. (35) of Appendix G, we carefully constructed a convergent series based on the collected samples to achieve an analyzable intermediate step for our proof of Theorem 6. More specifically, this special series is used to bound the estimation error of the updated occupancy measure and the estimated loss, which are functions of the estimated transition function based on the samples collected from the whole super-episode, in step-2-i and step-2-iv in Appendix G, respectively.
>
> - **(b)** Due to our new design of the estimated loss (12) that is based on the samples from the whole super-episode, the concentration lemma for the loss based on the samples from only one episode in Appendix B.3 in Jin et al. (2020a) does not apply in our case. To resolve this difficulty, we established a super-episodic version of concentration in step-2-iii of our proof for Theorem 6 in Appendix G by relating the second-order moment of the estimated loss that we design to that of the true loss. This is also a key new component in our extension. Without such a new component (e.g., directly applying the concentration lemma from Jin et al. (2020a)), there will be an additional factor $\tau$ in the first two terms of the upper bound in Eq. (40), which will finally lead to a $\\tilde{O}(T^{3/4})$ regret guarantee that is significantly larger than our current $\\tilde{O}(T^{2/3})$ regret guarantee.
>
> - **(c)** Due to the super-episode used in our SEEDS-UT algorithm, the original proof in Jin et al. (2020a) based on the loss in each episode does not apply any more. Although this difficulty can be resolved by our Lemma 7 when the transition function is known (as we discussed in A5), when extending to unknown transitions, Lemma 7 does not apply directly. This is because the occupancy measures on the left-hand-side of (19) in Lemma 7 is for different episodes of a super-episode. When the transition function is unknown, we have to make sure that these occupancy measures are the same. To resolve this issue, in Lemma 8 for this general case, we relate the updated occupancy measure based on the state-action-state triple for each episode to the true occupancy measure for the super-episode when the transition function is unknown. In this way, we can get an unknown-transition version of Lemma 7, which relates the regret bound in each episode to the final regret across super-episodes.
>
> To address this comment, in the revised paper, we have clarified further our new technical developments when extending known transitions to unknown transitions in Sec. 6 and in our proof of Theorem 6.
>
> Reference:
> Chi Jin, Tiancheng Jin, Haipeng Luo, Suvrit Sra, and Tiancheng Yu. Learning adversarial markov decision processes with bandit feedback and unknown transition. In International Conference on Machine Learning, pp. 4860–4869. PMLR, 2020a.
>
>
>
> Finally, we thank the reviewer again for the helpful comments and suggestions for our work. If our response resolves your concerns to a satisfactory level, we kindly ask the reviewer to consider raising the rating of our work. Certainly, we are more than happy to address any further questions that you may have during the discussion period.

---

> > ### Comment · Reviewer_y6Wb · 2022-12-01
> > **Reply to authors**
> >
> > Thank the authors for the detailed reply.
> >
> > For my statement that “the definition is unreasonable”, I mostly compare it with previous related works. I agree that this could happen in some cases. Considering the topic of this paper, such definition mismatches with previous works. For example, the seminal work [1] considers an adversarial MAB setting but with the definition of switch costs defined on pulled actions.
> >
> > Also, the authors might misunderstand the setting and lower bound mentioned in Shi et al. [2]. They consider a setting of adversarial MAB with switching costs and FULL-FEEDBACK costs, and their lower bound relies heavily on this assumption. Note that in the second paragraph of Section 3.2.1 (page 5), they construct an additional term and mention that “This additional noise is critical because our online algorithm $\pi$ can use costly full-feedback”. Since this paper does not consider paying full-feedback costs, the previous lower bound (though the switch costs are also defined on pulled actions) in [2] does not actually apply to this setting.
> >
> > Considering other issues like novelty, I would prefer to keep my score.
> >
> > Best,
> >
> > [1] Dekel, O., Ding, J., Koren, T., & Peres, Y. (2014, May). Bandits with switching costs: T 2/3 regret. In Proceedings of the forty-sixth annual ACM symposium on Theory of computing (pp. 459-467).
> >
> > [2] Shi, M., Lin, X., & Jiao, L. (2022, October). Power-of-2-arms for bandit learning with switching costs. In Proceedings of the Twenty-Third International Symposium on Theory, Algorithmic Foundations, and Protocol Design for Mobile Networks and Mobile Computing (pp. 131-140).

---

> > > ### Author Response · Authors · 2022-12-02
> > > **Response to Reviewer y6Wb**
> > >
> > > We thank the reviewer for the further feedback. Please see our response below.
> > >
> > > **Q1 [definition of the switching cost]:** For my statement that “the definition is unreasonable”, I mostly compare it with previous related works. I agree that this could happen in some cases. Considering the topic of this paper, such definition mismatches with previous works. For example, the seminal work (Dekel et al., 2014) considers an adversarial MAB setting but with the definition of switch costs defined on pulled actions.
> > >
> > > **A1:** We respectfully disagree with this comment. The reviewer refers to the switching cost in **MAB**, which is for changing actions. However, our paper studies **MDP** settings, and the standard definition of the switching cost for MDP, as in a well-established line of research such as Bai et al. (2019), Zhang et al. (2020), Wang et al. (2021), and Qiao et al. (2022), is exactly the same as that used in our paper.
> > >
> > > References:
> > >
> > > Ofer Dekel, Jian Ding, Tomer Koren, and Yuval Peres. Bandits with switching costs: $T^{2/3}$ regret. In Proceedings of the forty-sixth annual ACM symposium on Theory of computing, pp. 459–467, 2014.
> > >
> > > Yu Bai, Tengyang Xie, Nan Jiang, and Yu-Xiang Wang. Provably efficient q-learning with low switching cost. Advances in Neural Information Processing Systems, 32, 2019.
> > >
> > > Zihan Zhang, Yuan Zhou, and Xiangyang Ji. Almost optimal model-free reinforcement learning via reference-advantage decomposition. Advances in Neural Information Processing Systems, 33: 15198–15207, 2020.
> > >
> > > Tianhao Wang, Dongruo Zhou, and Quanquan Gu. Provably efficient reinforcement learning with linear function approximation under adaptivity constraints. Advances in Neural Information Processing Systems, 34:13524–13536, 2021.
> > >
> > > Dan Qiao, Ming Yin, Ming Min, and Yu-Xiang Wang. Sample-efficient reinforcement learning with $\log \log(t)$ switching cost. arXiv preprint arXiv:2202.06385, 2022.
> > >
> > > **Q2 [lower bound]:** Also, the authors might misunderstand the setting and lower bound mentioned in Shi et al. (2022). They consider a setting of adversarial MAB with switching costs and FULL-FEEDBACK costs, and their lower bound relies heavily on this assumption. Note that in the second paragraph of Section 3.2.1 (page 5), they construct an additional term and mention that “This additional noise is critical because our online algorithm $\pi$ can use costly full-feedback”. Since this paper does not consider paying full-feedback costs, the previous lower bound (though the switch costs are also defined on pulled actions) in Shi et al. (2022) does not actually apply to this setting.
> > >
> > > **A2:** We respectfully disagree with this comment. Note that the lower bound in Shi et al. (2022) holds for two cases: (i) the algorithms use full feedback and pay a cost for the full feedback, and (ii) the algorithms do not use full feedback (i.e., use only bandit feedback) and do not pay any feedback cost. Case (ii) is our case. Therefore, clearly the lower bound from Shi et al. (2022) works for our case.
> > >
> > > More specifically, the lower bound in Shi et al. (2022) is valid for the set $\\{\pi\\}$ of algorithms with full-feedback costs. Thus, the algorithms that choose to **never** use full feedback (which is exactly case (ii) above), i.e., $z(t)=0$ for all time $t$ in Eq. (1) in Shi et al. (2022), also belong to this set $\\{\pi\\}$ of algorithms (for which the lower bound in Shi et al. (2022) was constructed). Correspondingly, in the lower-bound proof in the technical report of Shi. et al (2022), this is the case when the number of times $N^{\text{ck}}$ of using costly full-feedback is $0$ in Eq. (60), and their lower bound still holds. Therefore, their lower bound also holds for the algorithms that do not use full feedback (i.e., use only bandit feedback) and do not pay any feedback cost.
> > >
> > > Reference:
> > > Ming Shi, Xiaojun Lin, and Lei Jiao. Power-of-2-arms for bandit learning with switching costs. In Proceedings of the Twenty-Third International Symposium on Theory, Algorithmic Foundations, and Protocol Design for Mobile Networks and Mobile Computing, pp. 131–140, 2022.
> > >
> > > **Q3:** Considering other issues like novelty, I would prefer to keep my score.
> > >
> > > **A3:** While we respect the reviewer’s current decision to keep the score, we kindly ask the reviewer to provide some details about why this is the case, at least be more specific about which novelty issues that the reviewer refers to, given that our first-round response has provided detailed answers to the reviewer’s novelty questions. More specifically, we have explained in detail in A5 (with (a) and (b) aspects) and A6 (with (a), (b), and (c) aspects) about the novel developments in this paper.

---

> > > > ### Comment · Reviewer_y6Wb · 2022-12-02
> > > > **Reply to authors**
> > > >
> > > > 1. These works the authors cited indeed study MDPs with switch costs. But all of them study “stochastic” MDPs with “deterministic” policies. More specifically, Bai et al. (2019) mentioned that “we focus on deterministic policies in this paper” in the second paragraph of Section 2 (page 3). Zhang et al. (2020) mentioned that “In this work, we mainly consider deterministic policies since the optimal value function can be achieved by a deterministic policy” in footnote 3 (page 3). Wang et al. (2021) clearly state that their policies are deterministic mappings (page 2). Qiao et al. (2022) mentioned that “our algorithm also uses deterministic policies only” in the last paragraph of section 2 (page 7).
> > > >
> > > > The paper I cited indeed studies bandit setting with switch costs, but it studies an “adversarial” setting, similar to this submission. Though an adversarial setting needs random policies, the switching costs are still defined on “realized actions”. Similar to the bandit case, the extension of the switching costs in adversarial MDPs would be natural if it is defined on “realized policies”, not underlying random policies. As I have mentioned in the original review, such a definition on random policies has a clear drawback that it neglects the real distances between distributions, while close distributions should not be treated very differently.
> > > >
> > > > 2. The lower bound in Shi et al. (2022) is also derived under the switch costs of “realized actions”. A change of pulled actions does not necessarily mean a change of random policies (where I don’t agree with the authors’ previous reply A2) and a change of random policies does not necessarily mean a change of pulled actions. Therefore, their lower bound could not simply apply in this setting.
> > > >
> > > > I wonder why the authors rely on the work Shi et al. (2022) which has the additional full-back costs, but not the work of Dekel et al. (2014) which has a closer setting with this submission.
> > > >
> > > > 3. For novelty, the proof process of Jin et al. (2020a) can be easily adjusted to this setting with the same results. First note that to analyze under switch costs, it is common to divide the total rounds into equal phases, which corresponds to “super-episodes” in this submission. Then the update of the occupancy measure only happens between super-episodes, thus we can treat the “super-episodes” as the episodes in Jin et al. (2020a) to get all the results. Specifically, the “episode” in the proof of a key B1 term in Lemma 4 of Jin et al. (2020a) can be replaced with the “super-episode” to get Eq. (35) in this submission; the “episode” in Lemma 12 of Jin et al. (2020a) can be replaced with the “super-episode” to get Eq. (40) in this submission; the “episode” in Lemma 14 of Jin et al. (2020a) can be replaced with the “super-episode” to get Step-2-iv in this submission. These three terms are the main differences while the techniques follow largely from the previous work. Thus I could not agree with the novelty of this submission.
> > > >
> > > > I guess the authors adopt such a definition of switch costs mainly because of the borrowed techniques, which study the adversarial MDP setting with natural random policies.

---

> > > > > ### Author Response · Authors · 2022-12-04
> > > > > **Response to Reviewer y6Wb (part2)**
> > > > >
> > > > > - **(a) The proof of Lemma 4 in Jin et al. (2020a) will not yield correct Eq. (35).** Due to the delayed switching of our SEEDS-UT algorithm (i.e., Algorithm 2 for the unknown transition case), our algorithm collects samples from an entire super-episode to estimate the transition function, and hence each state-action pair could be visited **multiple** times, and such visitations are **random**. In contrast, in Jin et al. (2020a), each state-action pair is assumed to have been visited at most once in an episode, which is required in many steps (e.g., Lemma 4 and Lemma 10) of their proof. Thus, the proof from Jin et al. (2020a) is not applicable in our case. Specifically, this new difficulty results in a different analysis that we developed for bounding the estimation error of the updated occupancy measure in step-2-i, i.e., the first term on the right-hand-side of (26). Notice that the difference between the estimated occupancy measure and the true occupancy measure in our case now depends on the samples from the entire super-episode (please refer to $q$, $\hat{q}$, and $\tilde{\epsilon}$ in Eq. (30)). Because of the reason above, simply considering the super-episode as an episode in Lemma 4 in Jin et al. (2020a) is clearly incorrect here. To resolve this difficulty, in Eq. (35) of Appendix G, we carefully constructed a series of terms based on the collected samples to achieve an analyzable intermediate step for our proof of Theorem 6.
> > > > >
> > > > > - **(b) Lemma 12 of Jin et al. (2020a) would yield a worse regret-bound in Eq. (40).** Due to our new design of the estimated loss (12) that is based on the samples from the entire super-episode, the concentration lemma for the loss based on the samples from only one episode in Appendix B.3 in Jin et al. (2020a) does not apply in our case. To resolve this difficulty, we established a super-episodic version of concentration in step-2-iii of our proof for Theorem 6 in Appendix G by relating the second-order moment of the estimated loss that we design to the super-episode length $\tau$ that need to be tuned and the true loss. This is also a key new component in our extension. Without such a new component (e.g., directly applying the concentration lemma and Lemma 12 from Jin et al. (2020a)), there will be an additional factor in the first two terms of the upper bound in Eq. (40), which will finally lead to a $O(T^{3/4})$ regret guarantee that is significantly larger than our current $O(T^{2/3})$ regret guarantee.
> > > > >
> > > > > - **(c) The proof of Lemma 14 of Jin et al. (2020a) does not apply to our Step-2-iv.** Simply replacing episodes by super-episode in Lemma 14 of Jin et al. (2020a) will be problematic in our case. One obvious problem is that their proof of Lemma 14 largely relies on Lemma 11, which further relies on the indicator function $I_{t,x,a}$ that they defined for whether a state-action pair $(x,a)$ is visited in a specific layer of the episode $t$. However, when replacing episodes by super-episodes, such an indicator function obviously does not make sense any more, since in a super-episode, the state-action pair could be visited in randomly multiple different episodes. Another problem is due to the different form of the estimated loss that we design in Eq. (12). Because of this, the proof of Lemma 11 from Eq. (17) to Eq. (18) and the last inequality in the proof of Lemma 14 in Jin et al. (2020a) do not work any more. Notice that our proof in Step-2-iv (e.g., Eq. (41) and the first inequality in Eq. (42)) is based on our new developments that we mentioned in (a) and (b) above and the technique elaborated in the following paragraph, which are clearly different from that of Lemma 14 in Jin et al. (2020a).\
> > > > > Due to the super-episode used in our SEEDS-UT algorithm, the original proof in Jin et al. (2020a) based on the loss in each episode does not apply any more. Although this difficulty can be resolved by our Lemma 7 when the transition function is known, when extending to unknown transitions, Lemma 7 does not apply directly. This is because the occupancy measures on the left-hand-side of (19) in Lemma 7 is for different episodes of a super-episode. When the transition function is unknown, we have to additionally make sure whether these occupancy measures are still the same or not. To resolve this issue, in Lemma 8 for this general case, we relate the updated occupancy measure based on the state-action-state triple for each episode to the true occupancy measure for the super-episode when the transition function is unknown. In this way, we can get an unknown-transition version of Lemma 7, which helps to relate the regret bound in each episode to the final regret across super-episodes.

---

> > > > > > ### Comment · Reviewer_y6Wb · 2022-12-06
> > > > > > **Reply to the authors**
> > > > > >
> > > > > > Thank the authors for the detailed reply.
> > > > > >
> > > > > > Perhaps I have misunderstood your setting since it is stated in the paper that the policy will draw actions by Eq.(5) and the later proofs are similar to the previous one. I have rechecked the proof, which seems to work under the deterministic "realized" policy within a super-episode. But since it is not clearly stated in the paper, can you confirm that your setting, algorithm, and proofs actually are processed by deterministic policies within "super-episodes"?
> > > > > >
> > > > > > If it is the case, then the setting makes sense to me and the lower bound could also apply.
> > > > > >
> > > > > > For the proof techniques, I agree that beyond simple replacement, there is still some preprocessing like dealing with multiple times of visitation of state-action pairs. I still feel that the techniques are very similar to the previous work. But if the setting has no problem, I think it is not a fundamental issue and I would like to increase my score to 6.
> > > > > >
> > > > > > Besides, there is a minor typo in the last inequality on page 27. In the LHS of this inequality, perhaps there is an additional $\sum_{u,s,a}$. In the RHS of this inequality, perhaps there is no $\eta$.
> > > > > >
> > > > > > Best,

---

> > > > > > > ### Author Response · Authors · 2022-12-06
> > > > > > > **Response to Reviewer y6Wb**
> > > > > > >
> > > > > > > We thank the reviewer for the reply and increasing the score of our work. Just a friendly reminder, could you please go to the recommendation section and change the score to 6, as indicated in your reply?
> > > > > > >
> > > > > > > We confirm that our setting, algorithms, and proofs are processed by deterministic policies within “super-episodes”. In the future version of this paper, we will make this clearer, and correct the typos that the reviewer pointed out. Thank you for the suggestions.

---

> > > > > ### Author Response · Authors · 2022-12-04
> > > > > **Response to Reviewer y6Wb (part1)**
> > > > >
> > > > > We thank the reviewer for the further feedback. Please see our response below.
> > > > >
> > > > > **Q1:** These works the authors cited indeed study MDPs with switch costs. But all of them study “stochastic” MDPs with “deterministic” policies. … The paper I cited indeed studies bandit setting with switch costs, but it studies an “adversarial” setting, similar to this submission. Though an adversarial setting needs random policies, the switching costs are still defined on “realized actions”. Similar to the bandit case, the extension of the switching costs in adversarial MDPs would be natural if it is defined on “realized policies”, not underlying random policies. As I have mentioned in the original review, such a definition on random policies has a clear drawback that it neglects the real distances between distributions, while close distributions should not be treated very differently.
> > > > >
> > > > > **A1:** The reviewer seems to have misunderstood our problem setting. We clarify that in this paper, we consider **deterministic** policies in each episode, not random policies. Please see the second paragraph in the problem formulation section, i.e., Sec. 3. Hence, we follow the standard definition of switching costs for deterministic policies in the MDP setting.
> > > > >
> > > > > Please note that although we consider only deterministic policies in each episode, since the decisions across episodes are random, the overall policy is still random, which is consistent with the general understanding that the policy for adversarial MDPs could be random.
> > > > >
> > > > > **Q2:** The lower bound in Shi et al. (2022) is also derived under the switch costs of “realized actions”. A change of pulled actions does not necessarily mean a change of random policies (where I don’t agree with the authors’ previous reply A2) and a change of random policies does not necessarily mean a change of pulled actions. Therefore, their lower bound could not simply apply in this setting. I wonder why the authors rely on the work Shi et al. (2022) which has the additional full-back costs, but not the work of Dekel et al. (2014) which has a closer setting with this submission.
> > > > >
> > > > > **A2:** As we explained in A1, in this paper, we consider only **deterministic** policies in each episode. Thus, a change of actions must indicate a change of policies. Therefore, as we mentioned in A2 of our previous response, the cost based on changing actions in Shi et al. (2022) will serve as a lower bound on the cost in our case.
> > > > >
> > > > > The reason that we did not use Dekel et al. (2014) for our lower bound development is because Dekel et al. (2014) does not characterize the dependency of the regret on the switching-cost coefficient and the range of losses. But the lower bound in Shi et al. (2022) further captures the dependency of the regret on those parameters in addition to $T$. Such a characterization enables us to develop a refined lower bound for MDP settings in our paper to capture the dependency of the regret on the parameters $H$, $S$, $A$ and $\beta$ (which are important for MDPs) in addition to $T$. In this way, we were able to show that our upper bound matches the lower bound in all of these parameters when the MDP is known, and in all except $H$ when the MDP is unknown.
> > > > >
> > > > > **Q3:** For novelty, the proof process of Jin et al. (2020a) can be easily adjusted to this setting with the same results. First note that to analyze under switch costs, it is common to divide the total rounds into equal phases, which corresponds to “super-episodes” in this submission. Then the update of the occupancy measure only happens between super-episodes, thus we can treat the “super-episodes” as the episodes in Jin et al. (2020a) to get all the results. Specifically, the “episode” in the proof of a key B1 term in Lemma 4 of Jin et al. (2020a) can be replaced with the “super-episode” to get Eq. (35) in this submission; the “episode” in Lemma 12 of Jin et al. (2020a) can be replaced with the “super-episode” to get Eq. (40) in this submission; the “episode” in Lemma 14 of Jin et al. (2020a) can be replaced with the “super-episode” to get Step-2-iv in this submission. These three terms are the main differences while the techniques follow largely from the previous work. Thus I could not agree with the novelty of this submission.
> > > > >
> > > > > **A3:** The reviewer appears to have overlooked our previous response A6 on our novel developments beyond Jin et al. (2020a). Below, we clarify the points in the context of the reviewer’s comments above to call the reviewer’s attention. Briefly speaking, although the policy over each super-episode in our case does not change, the estimation of the transition function as well as the occupancy measures is very different from that of the single episode in Jin et al. (2020a). Hence, just treating super-episodes as episodes in Jin et al. (2020a) will not even be correct.

---

> ### Author Response · Authors · 2022-11-15
> **Response to Reviewer y6Wb (part 2)**
>
> **Q4 [Question 2]:** Can you propose a more reasonable way to measure the difference between the two policy distributions and prove an upper bound?
>
> **A4:** There could be various metrics available to measure the difference between the two policy distributions, such as KL divergence, total variation, Wasserstein distance, etc. Since the policies are known by the agent, these metrics on the differences of the policies can be evaluated by the Monte Carlo method numerically.
>
> Moreover, if by “prove an upper bound”, the reviewer means “the regret upper bound”, then we may be able to develop a regret upper bound as follows. For example, for the offline approach suggested by the reviewer, building on prior works on online learning and off-policy evaluations, it may be possible to develop an upper bound on the regret for specific metrics. We also expect some challenges along this direction, e.g., how to set the threshold on how large the change of policies should trigger the actual online execution of the new policy, and how to balance the off-policy bias and switching cost, etc. Nevertheless, we believe this will be an interesting future problem worth further effort.
>
> **Q5 [Weakness 3]:** I did not see too many difficulties in the regret analysis process of Theorem 4. It seems that the proof process of Theorem 4 is a standard online mirror descent (OMD) proof process, except that super-episodes replace episodes. It is worth describing the non-trivial tricks of the proof process.
>
> **A5:** We respectfully disagree that the proof of Theorem 4 is a standard online mirror descent proof process, by simply replacing episodes by super-episodes. Note that within each super-episode (unlike episode) here, the loss function changes (possibly in an adversarial manner) from one episode to another, and each state-action pair could be visited multiple times in random episodes. These pose several new challenges that necessarily require new techniques. To resolve these challenges, our proof of Theorem 4 first upper-bounds the loss regret based on the correlated loss feedback in a super-episode and then relates these upper bounds across all super-episodes to the final regret. Both of these two steps and our new developments therein are different from the standard online mirror descent. Specifically, to achieve these two steps, our proof of Theorem 4 includes the following two new key developments.
>
> - **(a)** Note that, in Eq. (8), we designed a new estimator to estimate the loss function over the entire super-episode. Then, to upper-bound the loss regret based on the correlated loss feedback in a super-episode, a critical step in our proof of Theorem 4 is to show that such a new loss estimator is unbiased (as in the proof of Lemma 3 in Appendix C). This further requires new techniques to handle the random visitations of the state-action pairs in a super-episode. To this end, we established a novel property: under our new design of the estimated loss, summing over all possible sets of the random episodes where the state-action pair was visited is equivalent to summing over all deterministic episodes from the beginning to the end of a super-episode.
>
> - **(b)** To relate the above upper bounds across all super-episodes to the final regret, we established another important property in Lemma 7. Lemma 7 transfers the original regret formulation to a form based on the losses from the entire super-episode. In this way, we can relate the regret upper-bound in each episode to the final regret, which is critical to develop the overall regret bound in Theorem 4.
>
> As the reviewer suggested, in the revised paper, we have added more discussions about these non-trivial developments after Theorem 4 and in our proof of Theorem 4.

---

> ### Author Response · Authors · 2022-11-15
> **Response to Reviewer y6Wb (part 1)**
>
> We thank the reviewer for providing the helpful review! We have addressed the reviewer’s helpful comments and modified the paper accordingly. Please note that in the revised paper, we highlighted our changes by blue-colored texts in both the main body and the appendices of the paper.
>
> **Q1 [Weakness 1]:** The definition of the switching cost is unreasonable. In this paper, as long as the policy distribution changes slightly, there will be a switching cost. Imagine that if $\pi_{t+1}$ and $\pi_{t}$ are only slightly different (for example, a small KL divergence), we can fine-tune the policy. Then we can use offline evaluation methods such as importance sampling to evaluate the new policy, and there is no need to pay a high cost.
>
> **A1:** Thank you. Please note that this paper follows a well-established line of research, where the existing literature uses the same standard definition of “uniform switching cost” as in our paper, e.g., in Bai et al. (2019), Zhang et al. (2020), Wang et al. (2021), and Qiao et al. (2022). Such a definition does have justifications in practice. For example, in robotics (Kober et al., 2013), a slight change in policies could incur a large switching cost. However, we thank the reviewer for suggesting a possibly different notion of switching costs, which may depend on how large the policy changes, or only a large change of the policy incurs a switching cost. We anticipate that such more relaxed definitions can open up very interesting future directions and motivate different design guidelines, such as the offline approach suggested by the reviewer.
>
> References:
>
> Yu Bai, Tengyang Xie, Nan Jiang, and Yu-Xiang Wang. Provably efficient q-learning with low switching cost. Advances in Neural Information Processing Systems, 32, 2019.
>
> Zihan Zhang, Yuan Zhou, and Xiangyang Ji. Almost optimal model-free reinforcement learning via reference-advantage decomposition. Advances in Neural Information Processing Systems, 33: 15198–15207, 2020.
>
> Tianhao Wang, Dongruo Zhou, and Quanquan Gu. Provably efficient reinforcement learning with linear function approximation under adaptivity constraints. Advances in Neural Information Processing Systems, 34:13524–13536, 2021.
>
> Dan Qiao, Ming Yin, Ming Min, and Yu-Xiang Wang. Sample-efficient reinforcement learning with $\log \log(t)$ switching cost. arXiv preprint arXiv:2202.06385, 2022.
>
> Jens Kober, J Andrew Bagnell, and Jan Peters. Reinforcement learning in robotics: A survey. The International Journal of Robotics Research, 32(11):1238–1274, 2013.
>
> **Q2 [Weakness 2]:** The analysis of the lower bound in this paper is confusing. As stated in Weakness 1, the indicator function consider whether the policy distribution $\pi_{t+1}$ and $\pi_{t}$ are different, not $a_{t+1}$ and $a_{t}$ are the same. However, for adversarial bandits with switching costs, the indicator function considers whether $a_{t+1}$ and $a_{t}$ are different. These are two completely different measures, so the lower bound proved by Shi et al. (2022) cannot be used directly. Can you explain in detail why you can use the lower bound proved by Shi et al. (2022) directly?
>
> **A2:** Note that the change of the action $a_{t}$ indicates that the policy $\pi_{t}$ must change. Thus, the cost based on changing actions will serve as a lower bound on the cost here based on changing policies. From another perspective, note that the bandit case is a special case of our MDP case (when $S=H=1$). Thus, the lower bound on the bandits in Shi et al. (2022) will serve as a lower bound in our setting.
>
> However, the direct use of such a lower bound from bandits is not good enough for the MDP case that we study here. Thus, we further extended the approach in Shi et al. (2022) and constructed the worst-case state transitions to develop a better lower bound.
>
> In the revised paper, we have added more explanations at the beginning of Appendix A.
>
> Reference:
> Ming Shi, Xiaojun Lin, and Lei Jiao. Power-of-2-arms for bandit learning with switching costs. In Proceedings of the Twenty-Third International Symposium on Theory, Algorithmic Foundations, and Protocol Design for Mobile Networks and Mobile Computing, pp. 131–140, 2022.
>
> **Q3 [Question 1]:** Can you explain in detail why you can use the lower bound proved by Shi et al. (2022) directly?
>
> **A3:** Please see our answer A2.

---

> ### Author Response · Authors · 2022-11-29
> **Your feedback is important to us**
>
> Dear Reviewer y6Wb:
>
> This is a friendly reminder that we have submitted our response to your review comments two weeks ago, and we will appreciate very much if you could give us any feedback. In particular, our response explained in detail about your suggestion on the definition of switching costs, and we also clarified our novelty in the lower-bound development and regret analysis. If our response resolves your concerns, we kindly ask you to consider raising the rating of our work. We are also more than happy to answer your further questions. Thank you very much for your time and efforts!

---

### Decision · Program_Chairs · 2023-01-20

**Decision:**

Accept: notable-top-25%

**Justification For Why Not Higher Score:**

The topic is a niche for an ICLR audience but I think the story on the separation from the stochastic setting and the price of adaptivity is quite intriguing. It could be an interesting talk.

**Justification For Why Not Lower Score:**

The paper provides a clean solution that settles a theoretical problem of practical relevance.  The results are new and correct.

**Metareview: Summary, Strengths And Weaknesses:**

The paper studies RL with switching costs under the model of MDP with adversarial rewards. The goal is to design an algorithm that minimizes regret against the best fixed policy in hindsight. A switching cost --- defined as the number of times the chosen (deterministic) policy selected by the learner changes --- is added as part of the loss function (thus the regret).

The result shows that  T^{2/3}  regret is optimal with a lower bound and an algorithm that achieves this regret under both known and unknown transition settings.

The results are interesting as it separates the adversarial reward case with the stochastic reward case. Qiao et al. (2022) from ICML this year showed that one can obtain an $\tilde{O}(\sqrt{T})$ regret with $O(HSA \log \log T )$ switching cost and  \tilde{O}(T^{2/3}) regret with a constant $O(HSA)$ switching cost.   This paper shows that if the rewards are adversarial, there is a harsher tradeoff between learning and switching cost.

In terms of weaknesses, reviewers found that the proof techniques used in the paper mostly from existing work modulo minor modifications. There was also some confusion initially regarding whether the current paper works with deterministic or randomized policies which could affects the correctness of some of the proofs.  This got cleared up in the discussion and the reviewer who raised these questions is convinced that the proof will go through with deterministic policies.

**Note From Pc:**

if the above contains the word "oral" or "spotlight" please see: "oral" presentation means -> notable-top-5% and "spotlight" means -> notable-top-25%. As stated in our emails, we are disassociating presentation type from AC recommendations